# Decoupled neoantigen cross-presentation by dendritic cells limits anti-tumor immunity against tumors with heterogeneous neoantigen expression

**Kim Bich Nguyen[1,2], Malte Roerden[1], Christopher J Copeland[2], Coralie M Backlund[1,3], Nory G Klop-Packel[2], Tanaka Remba[1], Byungji Kim[1], Nishant K Singh[2], Michael E Birnbaum[2,3,4], Darrell J Irvine[2,3,4], Stefani Spranger[1,2,4,5]\***

[1]Koch Institute for Integrative Cancer Research, Massachusetts Institute of Technology, Cambridge, United States; [2]Department of Biology, Massachusetts Institute of Technology, Cambridge, United States; [3]Department of Biological Engineering, MIT, Cambridge, United States; [4]Ragon Institute of MGH, MIT and Harvard, Cambridge, United States; [5]Ludwig Center at MIT's Koch Institute for Integrative Cancer Research, Cambridge, United States

**\*For correspondence:**
spranger@mit.edu

**Abstract** Cancer immunotherapies, in particular checkpoint blockade immunotherapy (CBT), can induce control of cancer growth, with a fraction of patients experiencing durable responses. However, the majority of patients currently do not respond to CBT and the molecular determinants of resistance have not been fully elucidated. Mounting clinical evidence suggests that the clonal status of neoantigens (NeoAg) impacts the anti-tumor T cell response. High intratumor heterogeneity (ITH), where the majority of NeoAgs are expressed subclonally, is correlated with poor clinical response to CBT and poor infiltration with tumor-reactive T cells. However, the mechanism by which ITH blunts tumor-reactive T cells is unclear. We developed a transplantable murine lung cancer model to characterize the immune response against a defined set of NeoAgs expressed either clonally or subclonally to model low or high ITH, respectively. Here we show that clonal expression of a weakly immunogenic NeoAg with a relatively strong NeoAg increased the immunogenicity of tumors with low but not high ITH. Mechanistically we determined that clonal NeoAg expression allowed cross-presenting dendritic cells to acquire and present both NeoAgs. Dual NeoAg presentation by dendritic cells was associated with a more mature DC phenotype and a higher stimulatory capacity. These data suggest that clonal NeoAg expression can induce more potent anti-tumor responses due to more stimulatory dendritic cell:T cell interactions. Therapeutic vaccination targeting subclonally expressed NeoAgs could be used to boost anti-tumor T cell responses.

## Editor's evaluation

This valuable work explores the influence of intra tumor heterogeneity of neoepitopes within a cancer on the immune response leading to tumor control in vivo using a transplantable murine lung cancer model. It presents convincing evidence that immune responses against weak neoepitopes are enhanced when clonally expressed with strong neoepitopes, due to a more mature DC phenotype and a higher stimulatory capacity of DCs presenting both weak and strong neoepitopes. The work will be of interest to immunologists and cancer immunotherapists.

## Introduction

Engaging tumor-reactive immune responses has been an incredibly powerful tool in the fight against cancer (*Waldman et al., 2020*; *Esfahani et al., 2020*). Cytotoxic CD8[+] T cells recognize peptides on class I major histocompatibility complexes (MHCI) expressed on tumor cells, and following recognition mediate specific lysis of their target cell (*Stinchcombe et al., 2001*; *Isaaz et al., 1995*). While CD8[+] T cells can recognize many tumor-associated antigens, peptides specific to tumor cells are best suited to drive the most powerful anti-tumor responses (*Schietinger et al., 2008*; *Minati et al., 2020*). Amongst the tumor-specific antigens, the class of so-called neoantigens (NeoAg) is best understood thus far. NeoAgs are predominantly derived from non-synonymous mutations in highly expressed protein coding transcripts within the tumor cells (*Schumacher et al., 2019*). It has been shown that patients responding to checkpoint blockade immunotherapy (CBT) often experience an expansion in NeoAg-reactive T cells within tumor-infiltrating T cells, as well as in circulation (*van Rooij et al., 2013*; *Riaz et al., 2017*). Further, adoptive cell transfer of NeoAg-specific T cells may be beneficial (*Robbins et al., 2013*; *Verdegaal et al., 2016*; *Gros et al., 2014*) and vaccination can induce objective responses toward tumor-specific NeoAgs (*Carreno et al., 2015*; *Keskin et al., 2019*; *Johanns et al., 2019*; *Ott et al., 2017*).

The presence of CD8[+] T cells within the tumor microenvironment (TME) is established as a positive prognostic marker of response to CBT and overall survival (*van der Leun et al., 2020*). Over the past years, many studies have aimed to establish a correlation between the presence of NeoAgs and CD8[+] T cells within the tumor resulting in findings that increased NeoAg burden is positively associated with T cell infiltration in some cancers (*Rooney et al., 2015*). Despite enormous efforts, several independent reports suggest that NeoAg load alone cannot predict response to CBT (*Mauriello et al., 2019*; *McGrail et al., 2021*; *Samstein et al., 2019*; *Ghorani et al., 2018*). Of note, it seems the prognostic value of NeoAg burden depends on the baseline presence of a T cell infiltrate (*Mauriello et al., 2019*; *McGrail et al., 2021*). This can be best illustrated in cancer types with high mutational burden, such as melanoma, non-small cell lung cancer, and colon cancer. In those cancer types, a sizable proportion of patients lack a productive T cell infiltrate, despite an abundance in predicted NeoAgs (*Mauriello et al., 2019*; *McGrail et al., 2021*; *Spranger et al., 2016*). Past studies have indicated that alterations in tumor cell-intrinsic signaling pathways can mediate poor T cell infiltration, typically by means of poor T cell activation or poor T cell recruitment into the TME (*Nguyen and Spranger, 2020*; *Lawson et al., 2020*). However, these alterations do not account for all patients failing to respond to CBT while harboring high numbers of predicted NeoAgs. Recent studies suggest that intratumor heterogeneity (ITH), which might be highest in patients with high mutational burden, impacts the responsiveness to CBT (*McGranahan et al., 2016*). Clinical data suggest that clonal NeoAg expression is associated with response to anti-PD-1 CBT in a cohort of patients with NSCLC and with significantly increased overall survival in melanoma patients, following treatment with anti-CTLA-4 antibodies (*McGranahan et al., 2016*). In contrast, subclonal NeoAg expression in tumors with high ITH was found to be associated with poor CBT responses and poor CD8[+] T cell infiltration. These observations were confirmed in a transplantable mouse model using subclones derived from a UVB-irradiated murine melanoma cell line (*Wolf et al., 2019*).

While these initial studies strongly suggest that high ITH impairs the anti-tumor immune response, the mechanisms of how the anti-tumor immune response is impaired are still unknown. To interrogate the effect of ITH on anti-tumor T cell responses, we generated a syngeneic transplantable murine lung tumor model that enables us to precisely modulate the degree of ITH using naturally developed NeoAgs. Using two NeoAgs with different degrees of immunogenicity, we elucidated that responses against the weaker NeoAg were potentiated only in the clonal setting. This synergistic effect was established during T cell activation by cross-presenting conventional type I dendritic cells (cDC1), which acquired a more mature phenotype if they presented both antigens. Intriguingly, RNA-based vaccines targeting the weak NeoAg augmented immune responses in tumors with high ITH, highlighting the potential therapeutic value of targeting weakly immunogenic subclonal NeoAgs.

# Results

## Cancer cells expressing NeoAgs elicit diverse anti-tumor immune responses

Next-generation genome sequencing combined with MHCI binding prediction algorithms and in vivo validation have allowed for the identification of bona fide NeoAgs expressed in murine tumor lines including MC38, B16F10, and TRAMP-C1 (*Yadav et al., 2014*; *Castle et al., 2012*; *Matsushita et al., 2012*). Based on their reported immunogenicity, we selected candidate NeoAgs derived from mutated Adpgk, Aatf, and Cpne1 and immunized C57BL/6 mice with short peptides (8mer and 9mer) containing the mutations to validate their immunogenicity. Immunization with the mutant Adpgk peptide induced appreciable expansion of NeoAg-specific T cells while immunization with Cpne1 and Aatf peptides resulted in low or non-detectable T cell responses, respectively (*Figure 1A* and *Table 1*).

We next generated cell lines to assay anti-NeoAg responses in vivo by using KP1233, a lung adeno-carcinoma line derived from a $Kras^{G12D/+}Trp53^{-/-}$ mouse (*DuPage et al., 2009*). Because of the inherent cellular heterogeneity observed in many murine cell lines (*Ben-David et al., 2018*), we derived a stable subclone, referred to as KP6S, that grew similarly to the parental line in wildtype mice. The clonal KP6S cell line was used exclusively to generate all cell lines used in this study. To drive expression of specific NeoAgs, we expressed one or two NeoAgs linked at the C-terminus of the fluorescent protein mCherry followed by a barcode (*Figure 1B*). Subcutaneous implantation of the subclones expressing a single NeoAg (KP^Adpgk, KP^Aatf, KP^Cpne1) indicated that KP^Adpgk exhibited early tumor control before growing out while KP^Aatf and KP^Cpne1 cell lines grew out progressively (*Figure 1C*, *Figure 1—figure supplement 1A*). To confirm that the initial control observed against KP^Adpgk tumor cells was mediated by an adaptive immune response, we implanted all cell lines into $Rag2^{-/-}$ mice and observed that all three cell lines grew progressively with similar kinetics (*Figure 1D*, *Figure 1—figure supplement 1B*).

Naturally arising NeoAgs expressed in human cancer encompass both highly immunogenic and poorly immunogenic sequences. Thus, we chose the Adpgk, Aatf, and Cpne1 NeoAgs to capture the diversity of NeoAg-specific responses observed in humans (*Luksza et al., 2017*). Analysis of tumor-infiltrating T cells in KP^Adpgk and KP^Aatf showed that KP^Adpgk tumors had a greater degree of infiltration with CD8^+ T cells compared to KP^Aatf tumors (*Figure 1E and F*). Further, CD8^+ T cells in KP^Adpgk tumors were more activated based on CD44 staining (*Figure 1G*). Additionally, IFNγ ELISpot showed a greater peripheral expansion of NeoAg-specific T cells in mice implanted with KP^Adpgk tumors compared to KP^Aatf or KP^Cpne1 tumors (*Figure 1H*). Assessing the T cell infiltrate in KP^Cpne1 and KP^mCherry tumors revealed that Cpne1-expressing tumor cells were not highly immunogenic as neither CD8^+ T cell infiltration nor activation were significantly different between both tumors (*Figure 1—figure supplement 1C–E*). The immune responses observed corresponded with MHCI binding affinities predicted by NetMHC 4.0 (*Nielsen et al., 2003*; *Andreatta and Nielsen, 2016*), with the mutant Adpgk peptide predicted to have an IC_{50} of 4.29 nM while the other mutant peptides had IC_{50} values ranging from 90.16 nM (Aatf) to 182.34 nM (Cpne1) (*Figure 1—figure supplement 1F*). MHCI-stabilization assays also provided evidence that predicted binding affinities captured the range of peptide-MHCI (pMHCI) affinities for our selection of NeoAgs (*Figure 1—figure supplement 1G*). Thus, we established a model of transplantable syngeneic murine tumor lines that express NeoAgs with varying degrees of immunogenicity.

## Homogeneous expression of NeoAgs increases the immunogenicity of cancer cells

To assess the impact of different NeoAg expression patterns in tumors, we first generated a cell line that expressed both Adpgk and Aatf, hereafter termed KP-Het^Low (*Figure 2A*). To model heterogeneous NeoAg expression patterns (KP-Het^High), we inoculated C57BL/6 mice with a mixture of 50% KP^Aatf cells and 50% KP^Adpgk cells (*Figure 2A*). We implanted $1 \times 10^6$ cells of KP-Het^High and KP-Het^Low tumors into mice and observed drastically increased control of tumor outgrowth of KP-Het^Low tumors compared to single antigen-expressing tumors. In contrast, KP-Het^High grew progressively, with similar kinetics as observed for KP^Aatf (*Figure 2B*). In fact, quantitative PCR analysis of KP-Het^High tumors showed progressive outgrowth of the KP^Aatf subclone that completely dominated the tumor by day 14 post-implantation (*Figure 2—figure supplement 1A*). The control of KP-Het^Low tumors was

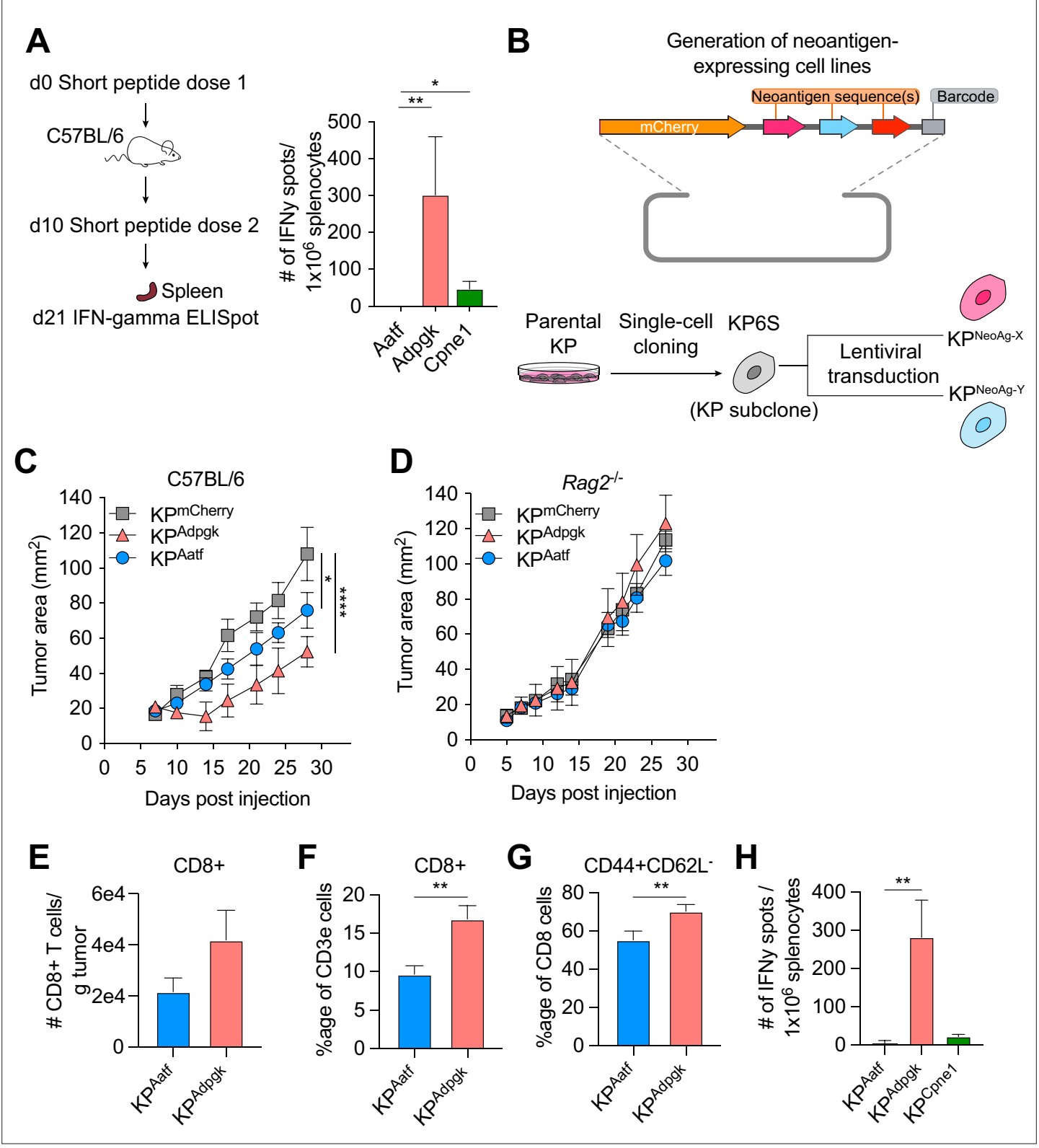

**Figure 1.** KP6S cell line engineered to express natural neoantigens (NeoAgs) elicit variable anti-tumor immune responses. (**A**) Mice were vaccinated with short peptides with cyclic-di-GMP as adjuvant. 10 µg of peptide was delivered subcutaneously (s.c.) at the base of the tail along with 25 µg of cyclic-di-GMP. An identical dose was delivered s.c. 10 days following the first dose and spleens were collected at day 21 for IFNγ ELISpot. Quantification of IFNγ-producing cells after restimulation from two independent experiments shown as mean ± SEM (n = 3 per group per experiment). (**B**) Schematic

*Figure 1 continued on next page*

*Figure 1 continued*

of the lentiviral construct used to transduce the KP6S subclone. (**C, D**) Mice were injected s.c. with 1 × 10^6 tumor cells in (**B**) WT mice or (**C**) *Rag2*^-/- mice. Representative data from one of two individual experiments are shown (n = 3 or 4 per group per experiment). Quantification of (**E**) absolute numbers of CD8+ TIL per gram tumor from six independent experiments (pooled n = 17 per group), (**F**) proportion of CD8+ TIL at day 9 or 10 after tumor implantation from eight independent experiments (pooled n = 23 per group), (**G**) proportion of CD44+CD62L- T_effector from eight independent experiments (pooled n = 23 per group), (**H**) IFNγ-producing cells restimulated 9 or 10 d after tumor implantation using ELISpot from two independent experiments (pooled n = 5 per group). *p<0.05, **p<0.01, ****p<0.0001; one-way ANOVA (Kruskal–Wallis) test in (**A**), two-way ANOVA (Tukey) in (**C, D**), Mann–Whitney *U* in (**E–H**). Data are shown as mean ± SEM.

The online version of this article includes the following source data and figure supplement(s) for figure 1:

**Source data 1.** Raw data for *Figure 1*.

**Figure supplement 1.** Characterization of the immunogenicity of an array of natural neoantigens (NeoAgs).

**Figure supplement 1—source data 1.** Raw data for *Figure 1—figure supplement 1*.

completely lost in *Rag2*^-/- mice (*Figure 2B*), indicating that tumor control was mediated by an adaptive immune response. We also found that the change in tumor cell composition of KP-Het^High tumors at later timepoints observed in wildtype mice was also absent in *Rag2*^-/- mutants. Instead, KP^Aatf and KP^Adpgk cells were maintained at nearly a 1:1 ratio (*Figure 2—figure supplement 1B*), providing evidence for immunoediting in this model. These data indicate that adaptive immune responses are capable of controlling tumors with homogeneous NeoAg expression, whereas tumors with heterogeneous NeoAg expression are characterized by immune editing and escape of non-immunogenic subclones.

To obtain insights into the kinetics of the tumor-reactive T cell response, we assayed NeoAg-specific T cells via IFNγ ELISpot on days 7, 10, and 14 post tumor implantation. The T cell response toward Adpgk was significantly greater in KP-Het^Low tumors compared to KP-Het^High tumors at day 7 and greater than both KP-Het^High and KP^Adpgk tumors at day 10 (*Figure 2C*). Strikingly, at days 7 and 10, the T cell response against Aatf was only detectable in mice implanted with KP-Het^Low tumors and was absent in mice bearing KP-Het^High or KP^Aatf tumors (*Figure 2D*). At day 14, the Aatf and Adpgk responses were similar in all tested conditions, suggesting mixed effects of tumor size, antigen availability, and loss of functional capacity of T cells over time (*Figure 2B–D*). The observed enhanced T cell response against a weakly immunogenic NeoAg is also observed when Cpne1 was co-expressed with Adpgk (*Figure 2—figure supplement 2A and B*). We considered the possibility that increasing NeoAg load in a cell could increase immunogenicity by expressing the two weakly immunogenic NeoAgs, Aatf and Cpne1, together. However, this provided no benefit to the Aatf response (*Figure 2—figure supplement 3*). These data suggest that homogeneous NeoAg expression patterns can increase the peripheral response against poorly immunogenic NeoAgs if they are paired in tandem with a stronger antigen.

We next assessed the relative contribution of each NeoAg-specific immune response to the superior control of KP-Het^Low tumors. Adoptively transferred CD8^+ T cells from KP-Het^Low-bearing donor mice slowed the growth of KP^Adpgk as well as KP^Aatf in *Rag2*^-/- mice (*Figure 2E*). In line with the weaker IFNγ ELISpot responses observed in KP-Het^High tumors, transfer of CD8^+ T cells from KP-Het^High-bearing donor mice was less beneficial, resulting in improved control of KP^Adpgk, but not of KP^Aatf. While adoptive cell transfer more effectively slowed the growth of KP^Adpgk tumors, this suggests that the Aatf-specific immune response contributes to the superior tumor control of KP-Het^Low tumors.

**Table 1.** Amino acid sequences of wildtype and NeoAg.

| Name | Wildtype | | | | Mutant | | |
| | Peptide sequence | Sequence position | Binding prediction | Predicted affinity (IC50) | Peptide sequence | Binding prediction | Predicted affinity (IC50) |
|---|---|---|---|---|---|---|---|
| Adpgk | ASMTNRELM | 298–307 | Db | 6.21 | ASMTN**M**ELM | Db | 4.29 |
| Aatf | MAPIDHTAM | 493–501 | Db | 297.14 | MAPIDH**T**TM | Db | 90.16 |
| Cpne1 | SSPDSLHYL | 298–307 | Db | 764.37 | SSP**Y**SLHYL | Db | 182.34 |

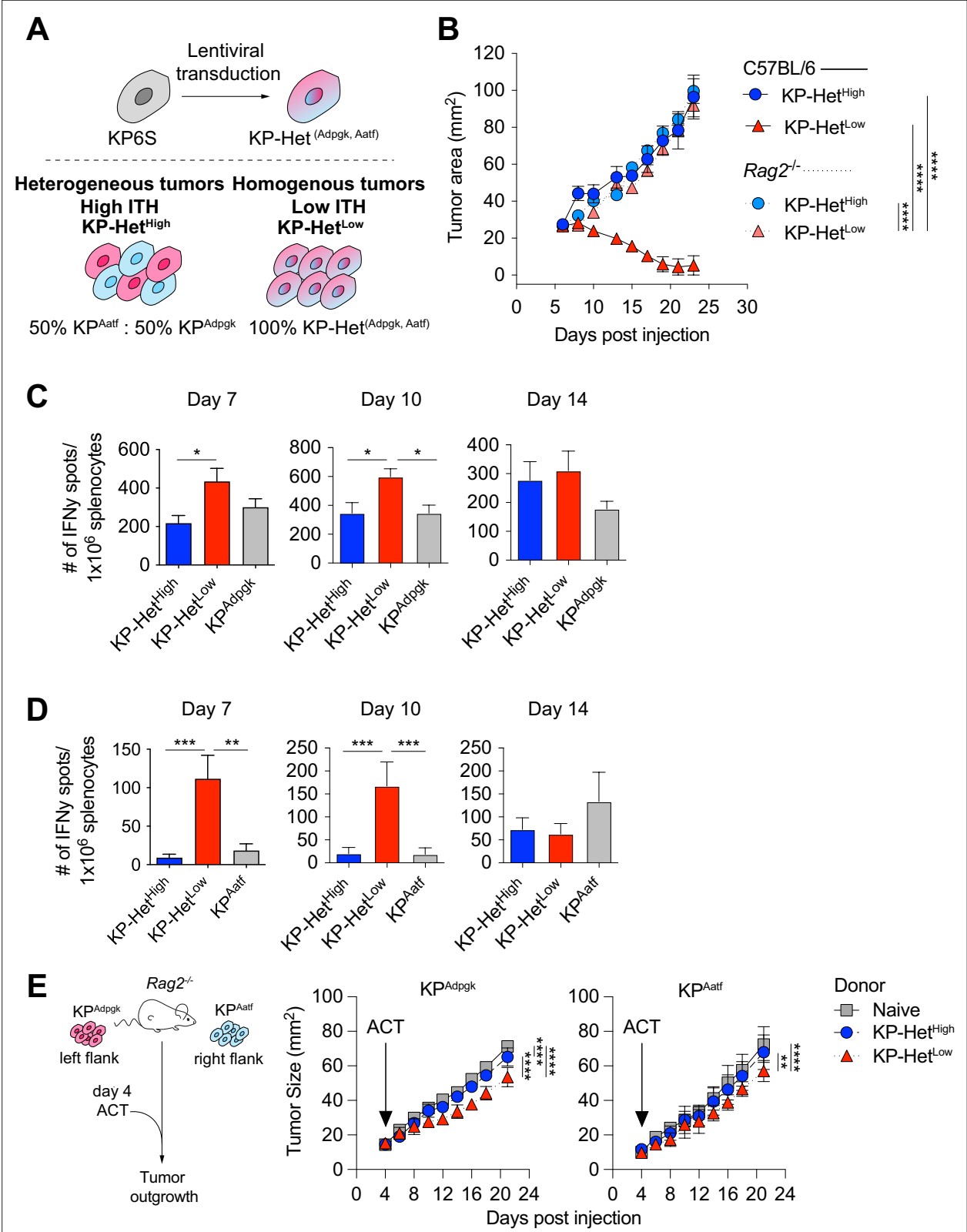

**Figure 2.** Tumors expressing a pair of neoantigens (NeoAgs) homogeneously have increased immunogenicity. (**A**) Schematic of the generation of tumors used in (**B**). (**B**) Tumor growth of KP-Het^High and KP-Het^Low in WT and *Rag2*^-/- mice. Representative data from one of two individual experiments are shown (n = 3 per group per experiment). (**C, D**) Splenocytes from tumor-bearing mice were used in an IFNγ ELISpot to determine the frequency of NeoAg-specific T cells in the periphery at days 7, 10, and 14 after tumor implantation. Quantification of the (**C**) Adpgk-specific response and (**D**) Aatf-

*Figure 2 continued on next page*

*Figure 2 continued*

specific response. Pooled data from five independent experiments for day 7 for single antigen tumors and six independent experiments for all other groups (n = 3–4 per group per experiment), four independent experiments for day 10 for single antigen tumors and five independent experiments for all other groups (n = 3 per group per experiment) and three independent experiments for day 14 (n = 3 per group per experiment) in (**C, D**). (**E**) Schematic and tumor growth of KP$^{Aatf}$ and KP$^{Adpgk}$ in *Rag2$^{-/-}$* mice after adoptive T cell transfer (ACT) from naïve or tumor-bearing mice on day 4 after tumor injection. Representative data from one of two individual experiments are shown (n = 4 per group per experiment). *p<0.05, ***p<0.001, ****p<0.0001; two-way ANOVA (Tukey) in (**B, E**), one-way ANOVA (Kruskal–Wallis followed by Dunn's multiple-comparisons test) in (**C, D**). Data are shown as mean ± SEM.

The online version of this article includes the following source data and figure supplement(s) for figure 2:

**Source data 1.** Raw data for *Figure 2*.

**Figure supplement 1.** KP-Het$^{High}$ tumors are immune edited.

**Figure supplement 1—source data 1.** Raw data for *Figure 2—figure supplement 1*.

**Figure supplement 2.** Expansion of neoantigen (NeoAg)-specific T cells directed against weak antigens occurs earlier when they are co-expressed with a stronger antigen.

**Figure supplement 2—source data 1.** Raw data for *Figure 2—figure supplement 2*.

**Figure supplement 3.** Increased Aatf-specific T cell expansion is not due to increased numbers of antigens expressed in KP-Het$^{Low}$ tumors.

**Figure supplement 3—source data 1.** Raw data for *Figure 2—figure supplement 3*.

## Batf3$^{+}$ dendritic cells are required for anti-tumor responses in KP-Het$^{Low}$ tumors

Given our observation that peripheral T cell responses against weak NeoAgs are enhanced early following tumor inoculation, we postulated that T cell activation of Aatf-reactive T cells in the lymph node might be different between mice bearing KP-Het$^{Low}$ and KP-Het$^{High}$ tumors. While it is established that cross-presenting cDC1 driven by the transcription factor Batf3 are critical for priming CD8$^{+}$-specific responses (*Hildner et al., 2008*; *Spranger et al., 2015*), recent work by us and others have also implicated additional cDC subsets (*Duong et al., 2022*) or compensatory development of Batf3-independent cDC1 (*Tussiwand et al., 2012*) in mediating anti-tumor immunity. We thus aimed to determine whether Batf3-dependent cDC1 were required for the increased immune control observed against KP-Het$^{Low}$ tumors. We implanted KP-Het$^{Low}$ tumor cells in wildtype, *Rag2$^{-/-}$* and *Batf3$^{-/-}$* mice, and observed a loss of tumor control in *Rag2$^{-/-}$* and *Batf3$^{-/-}$* mice (*Figure 3A*), indicating that cDC1 are required for the induction of effective T cell responses.

cDC1 can impact anti-tumor T cell responses during T cell activation in the tumor-draining lymph node (TdLN) or by facilitating recruitment to the tumor (*Spranger et al., 2015*). Since we observed differences in CD8$^{+}$ T cell infiltration between KP-Het$^{Low}$ and KP-Het$^{High}$, we first assessed the number of tumor-infiltrating cDC1. However, while we observed dynamic changes in the absolute numbers of cDC1 over time, no significant difference was found between the two tumor conditions (*Figure 3B*, *Figure 3—figure supplement 1*). To track cDC1 carrying tumor cell debris, we controlled for mCherry expression in all cell lines by assessing the fluorescent intensity using flow cytometry to ensure equal antigen and fluorophore expression (*Figure 3—figure supplement 2*). Assessing the number of tumor cell debris carrying mCherry$^{+}$ cDC1 in the TdLN further affirmed that the differences in T cell activation were not driven by a lack of migratory cDC1 bringing antigen to the TdLN as similar frequencies were detected between the two tumor conditions (*Figure 3C*, *Figure 3—figure supplement 1*). Analysis of the mCherry MFI amongst the mCherry$^{+}$ cDC1 similarly showed no significant difference between the KP-Het$^{Low}$ and KP-Het$^{High}$ conditions (*Figure 3D*), suggesting that neither cDC1 recruitment to the tumor, trafficking to the TdLN, nor amount of available antigen can explain the observed differences in T cell activation.

In the homogeneous KP-Het$^{Low}$ setting, it is conceivable that epitope spreading in response to a rapid and strong Adpgk-specific T cell response might lead to an increase in available Aatf antigen as killing of KP-Het$^{Low}$ cells would result in release of both Adpgk and Aatf-containing debris. This increase in antigen abundance could explain an increase in activation of Aatf-reactive T cells compared to the KP-Het$^{High}$ setting. To test whether antigen availability alone might explain the differences in T cell response, we inoculated mice with lethally irradiated tumor cells using single-antigen-expressing tumor cell lines (KP$^{Aatf}$ or KP$^{Adpgk}$), or the KP-Het$^{Low}$ and KP-Het$^{High}$ conditions (*Figure 3E*). To ensure robust responses, we recalled T cell responses with an equal mixture of purified Adpgk and Aatf short

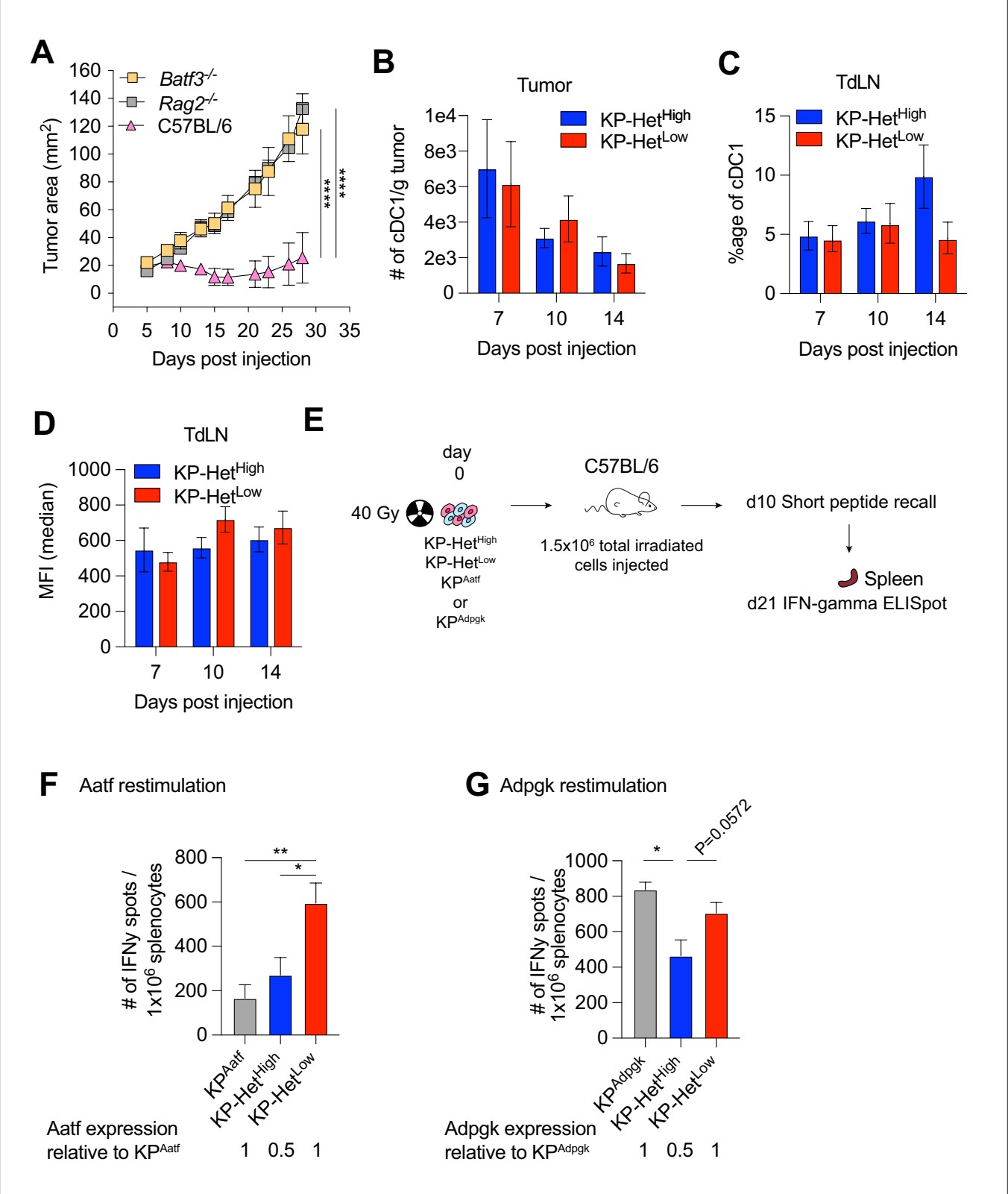

**Figure 3.** Cross-presenting dendritic cells mediate the increased immunogenicity of KP-Het^Low tumors. (**A**) Tumor growth of KP-Het^High tumor cells was implanted subcutaneously (s.c.) into *Batf3^-/-*, *Rag2^-/-* and WT mice. Representative data from three independent experiments (n = 5 per group per experiment). (**B**) Number of cDC1 in KP-Het^High and KP-Het^Low tumors on days 7, 10, and 14 after s.c. implantation. Pooled data from two independent experiments is shown (n = 3 per group per experiment). (**C**) Proportion of mCherry^+ cDC1 in tumor-draining lymph nodes. Pooled data from two

*Figure 3 continued on next page*

*Figure 3 continued*

independent experiments for days 7 and 10 and three independent experiments for day 14 is shown (n = 3 per group per experiment). (**D**) Median fluorescence intensity of the mCherry signal of cells from (**B**). (**E**) Experimental schematic for (**F, G**). Tumor cells were irradiated with 40 Gy and 1.5 × $10^6$ total irradiated cells were immediately s.c. injected into mice. A short peptide boost with both peptides and c-di-GMP as adjuvant was given 10 d after and administered s.c. at the base of the day. 21 days after the irradiated cell implantation, spleens were collected for ELISpot. (**F**) Peripheral Aatf-specific response. Pooled data from three independent experiments are shown (pooled n = 11 or 12 per group). (**G**) Peripheral Adpgk-specific response. Pooled data from one or three independent experiments are shown (n = 6 for KP$^{Adpgk}$ and pooled n = 12 for remaining groups). *p<0.05, **p<0.01, ****p<0.0001; two-way ANOVA (Tukey) in (**A**), Mann–Whitney $U$ for each time point between the two tumors was assessed in (**B–D**), one-way ANOVA (Kruskal–Wallis followed by Dunn's multiple-comparisons test) in (**F, G**). Data are shown as mean ± SEM.

The online version of this article includes the following source data and figure supplement(s) for figure 3:

**Source data 1.** Raw data for *Figure 3*.

**Figure supplement 1.** Gating strategy for cDC1.

**Figure supplement 1—source data 1.** Raw data for *Figure 3—figure supplement 1*.

**Figure supplement 2.** KP cell lines express similar levels of neoantigens (NeoAg).

**Figure supplement 2—source data 1.** Raw data for *Figure 3—figure supplement 2*.

peptides combined with cyclic-di-GMP as an adjuvant 10 d after the initial injection of irradiated tumor cells (*Figure 3E*). Then 11 d post recall, T cell responses were assessed using an IFNγ ELISpot assay (*Figure 3E*). Consistent with our previous observations, we observed that the Aatf-specific T cell response was dependent on the context of the NeoAg expression patterns, with greater expansion of Aatf-specific T cells in response to KP-Het$^{Low}$ tumor debris compared to either KP$^{Aatf}$ or KP-Het$^{High}$ tumor debris (*Figure 3F*). In contrast, we did observe that the Adpgk response was sensitive to lower antigen availability as mice injected with irradiated KP-Het$^{High}$ tumor cells, where only 50% of the cells express Adpgk, also exhibited a significantly reduced expansion of Adpgk-specific T cells in the periphery compared to KP$^{Adpgk}$ and KP-Het$^{Low}$, both tumors where all the cells express Adpgk (*Figure 3G*). This result is consistent with previous reports on the correlation between antigen availability and strength of T cell response, where providing less Adpgk debris resulted in a corresponding decrease in the Adpgk-specific response (*Bullock et al., 2003*; *Westcott et al., 2021*). In sum, we identified that NeoAg expression patterns are critical for priming responses against weak NeoAgs, while the antigen load impacts responses toward strong NeoAgs.

## NeoAg presentation on dendritic cells mirrors NeoAg expression patterns in the TME

It has been shown that the same dendritic cell can take up debris containing both MHCII- and MHCI-restricted epitopes, allowing the DC to interact with CD4$^+$ T cells for licensing to then activate a productive CD8$^+$ T cell response (*Ferris et al., 2020b*). Similarly, reports suggest that interactions between a DC and CD8$^+$ T cells can impact the maturation state of the DC (*Mailliard et al., 2002*; *Hernandez et al., 2007*). We thus considered the possibility that a strong MHCI epitope might act as a 'licensing' response to a weaker MHCI epitope when presented on the same DC. To test this notion, we regenerated the KP-Het$^{Low}$ cell lines by expressing Adpgk linked at the C-terminus of ZsGreen (ZsG) while Aatf maintained its expression with mCherry establishing KP-Het$^{Low(ZsG-Adpgk,Aatf)}$, and a corresponding KP$^{ZsG-Adpgk}$ as control (*Figure 4—figure supplement 1A*). We confirmed that these tumor cell lines recapitulated the previously observed outgrowth kinetics (*Figure 4—figure supplement 1B*). Given that we established the importance of cDC1 for T cell priming, we focused our analysis on this DC subset and used mCherry and ZsGreen as a readout for tumor cell debris engulfment and antigen presentation (*Figure 4A*). We first determined the proportion of single-fluorophore or double-fluorophore positive cDC1 in the TdLN at day 7 post tumor implantation and found that in KP-Het$^{High}$ tumors most of the cDC1 carrying detectable debris were either mCherry$^+$ or ZsGreen$^+$ (*Figure 4B*). In stark contrast, most tumor cell debris-positive cDC1 found in the TdLN-draining KP-Het$^{Low}$ tumors were double positive for both mCherry and ZsGreen (*Figure 4B*). Within the single positive cDC1 subset in both tumors, there was a bias toward ZsGreen$^+$ cells (*Figure 4C*), which could be attributed to the stability of this fluorescent protein (*Yi et al., 2022*).

Previous reports have indicated that costimulatory markers were upregulated on dendritic cells following 'licensing' interactions with CD4$^+$ but also CD8$^+$ T cells (*Mailliard et al., 2002*; *Hernandez*

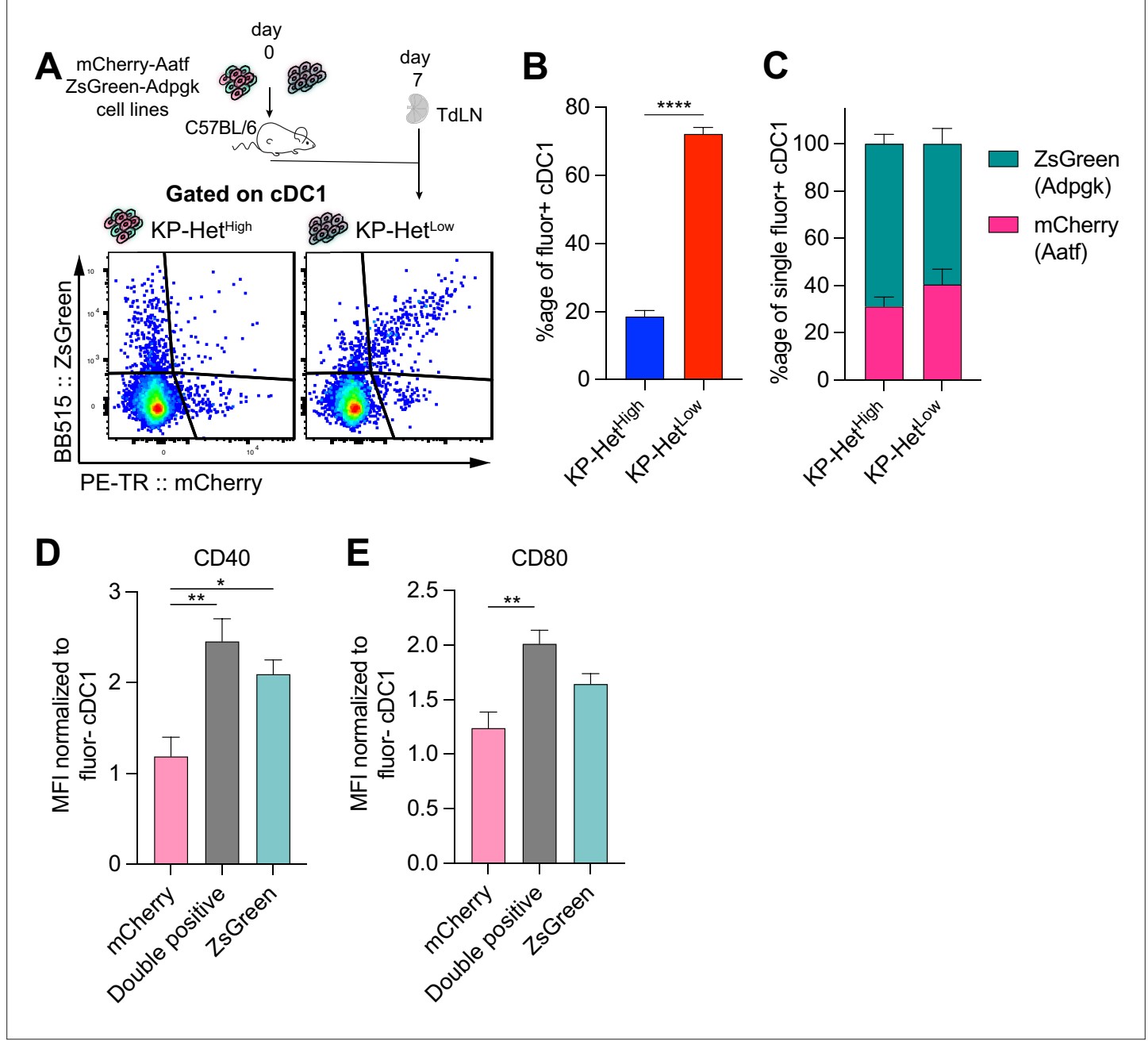

**Figure 4.** Antigen presentation on dendritic cells in the tumor-draining lymph node mirror antigen expression patterns in the tumor microenvironment (TME). (**A**) Experimental schematic for (**B, C**). KP-Het^High tumors were composed of KP^ZsG-Adpgk and KP^Aatf; KP-Het^Low tumors were composed of KP-Het^Low(ZsG-Adpgk,Aatf). (**B**) Quantification of the proportion of cDC1 that are double positive (mCherry+ZsGreen+) in tumors. (**C**) Proportion of mCherry+ or Zsgreen+ cDC1 in the single positive population. Pooled data from three independent experiments are shown (pooled n = 10 per group) for (**B, C**). (**D**) Normalized CD40 median fluorescence intensity for single-positive and double-positive populations. (**E**) Normalized CD80 median fluorescence intensity for the same sample populations in (**D**). Pooled data from three independent experiments are shown (pooled n = 13 per group) for (**D**) and (**E**). *p<0.05, **p<0.01; one-way ANOVA (Kruskal–Wallis followed by Dunn's multiple-comparisons test) in (**B–E**). Data are shown as mean ± SEM.

The online version of this article includes the following source data and figure supplement(s) for figure 4:

**Source data 1.** Raw data for *Figure 4*.

**Figure supplement 1.** Clonal expression of Adpgk and Aatf neoantigens (NeoAgs) results in increased immunogenicity regardless of linked or separate expression of NeoAgs in the same cell.

**Figure supplement 1—source data 1.** Raw data for *Figure 4—figure supplement 1*.

*et al., 2007*; *Carenza et al., 2019*; *Min et al., 2010*). We therefore assessed the expression of CD40 and CD80 on single or double fluorophore-positive cDC1 populations in the TdLN. Affirming our initial hypothesis, we observed significantly greater expression of the costimulatory molecules CD40 and CD80 in double-positive cDC1 compared to mCherry⁺ cDC1 that engulf only Aatf-containing debris (*Figure 4D and E*). CD40 expression was comparable between Adpgk-ZsGreen⁺ and double-positive cDC1, while CD80 was highly expressed on both these cDC1 populations (*Figure 4D and E*), suggesting that the Adpgk-specific T cell response might induce upregulation of co-stimulatory molecules. In sum, these findings suggest that the antigen-dependent interaction between cDC1 and Adpgk-specific T cells could result in increased activation ('licensing') of cDC1, and subsequently, an increased capacity of cDC1 to prime Aatf-reactive T cells, if the same cDC1 also presents the weaker NeoAg.

## Prophylactic RNA vaccination expands Aatf-specific T cells and increases response of heterogeneous tumors to CBT

Clinically, a high degree of ITH is associated with poor responses to CBT. To determine whether our established model system faithfully recapitulated resistance to therapy, we inoculated KP-Het^Low or KP-Het^High in C57BL/6 mice and treated mice with dual CBT, consisting of anti-CTLA4 and anti-PD-L1 antibodies on days 7, 10, 13, and 16. Consistent with clinical observations, our model showed that KP-Het^Low tumors could be fully controlled following therapy (*Figure 5A*). In contrast, KP-Het^High tumors, engineered to resemble tumor with high ITH, showed mixed responses (*Figure 5A*). Our gained insights into the mechanism of resistance in tumor with heterogeneous NeoAg expression suggest that increased tumor control in homogeneous tumor was associated with a more rapid and robust expansion of T cell responses toward weaker NeoAgs. Therefore, we aimed to determine whether prophylactic vaccination might increase tumor control of KP-Het^High tumors in the context of dual CBT. Mice were vaccinated with self-amplifying RNA (replicons) encased in a lipid nanoparticle administered intramuscularly (i.m.) and boosted i.m. 2 wk after the initial dose. Using this strategy, we induced a detectable Aatf-specific response 7 d post boost (*Figure 5B*), unlike short peptide vaccinations (*Figure 1A*). $1 \times 10^6$ KP-Het^High tumor cells were inoculated 7 d post boost and dual CBT was administered on days 7, 10, 13 and 16 post tumor implantation (*Figure 5C*). As seen before, KP-Het^High tumors showed mixed responses following CBT, ranging from progressive growth to stable disease (*Figure 5D*). Prophylactic vaccination alone only resulted in a modest reduction of tumor growth (*Figure 5D*). However, we observed a synergistic effect of the combination treatment of prophylactic vaccination and dual CBT, with two out of seven (29%) objective responses with one complete response and one stable disease (*Figure 5D*). Thus, we established proof of concept that targeting a subclonal, weakly immunogenic NeoAg could be a viable strategy to increase CBT response.

## Therapeutic RNA vaccination with CD40 agonism synergizes with CBT in KP-Het^High tumors

Given that the majority of NeoAgs emerge during tumor development and thus disqualify prophylactic vaccination, we next aimed to determine the utility of therapeutic vaccination with CBT as a therapeutic strategy. Four days after tumor implantation, mice were treated with RNA replicons and continually dosed every week thereafter (*Figure 6A*). Our analysis of the cDC1 compartment in tumors with clonal NeoAg expression further suggested that CD40:CD40 ligand interactions would enhance induction of Aatf-reactive effector T cell responses (*Figure 4D*). Furthermore, previous studies have shown that CD40 stimulation could induce strong systemic anti-tumor responses leading to regression of tumors in preclinical models (*van Mierlo et al., 2002*; *Sandin et al., 2014*), and it has also been shown to synergize with CBT (*Westcott et al., 2021*; *Morrison et al., 2020*). Thus, we combined the vaccination approach with a single dose of agonistic anti-CD40 antibody along with the first vaccine dose (*Figure 6A*). Dual CBT was administered intraperitoneally (i.p.) on days 7, 10, 13 and 16 post tumor implantation (*Figure 6A*).

Similar to CBT alone, vaccination alone or agonistic CD-40 antibody administration exhibited only a modest response characterized by slowing of tumor growth (*Figure 6B*). In stark contrast, the triple combination of CBT with therapeutic vaccination and a single dose of agonistic CD40 antibody induced complete tumor control in three of five (60%) mice and a significant delay in tumor growth in the remaining 40% of mice (*Figure 6C*). CD40 agonism with vaccination was able to induce a

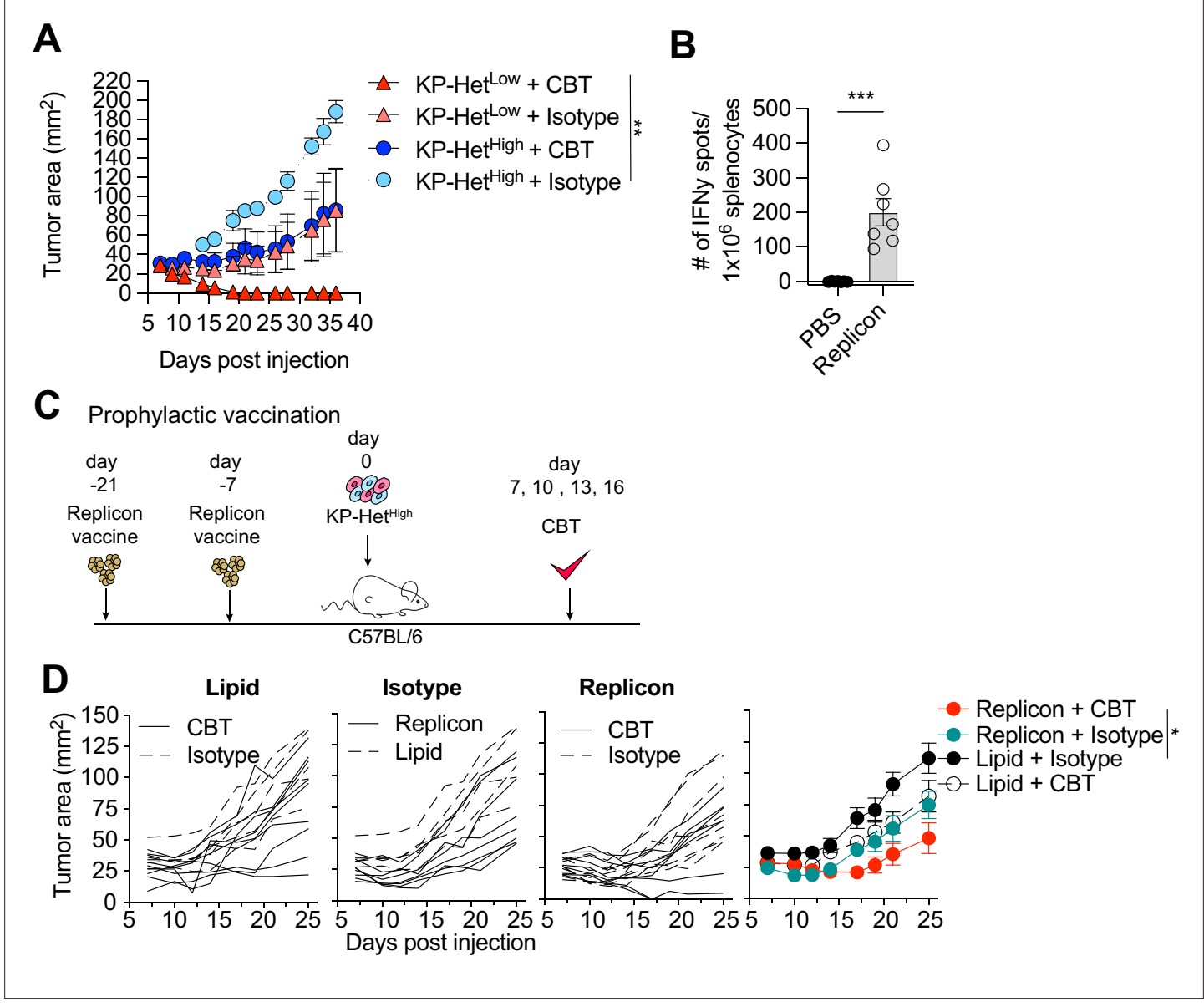

**Figure 5.** Prophylactic mRNA replicon vaccination increases response to checkpoint blockade therapy in KP-Het[High] tumors. (**A**) Tumor growth of KP-Het[High] and KP-Het[Low] WT mice treated with checkpoint blockade immunotherapy (CBT) or control. 100 µg of each antibody (CBT or isotype control) was administered intraperitoneally (i.p.) on days 7, 10, 13, and 16 after implantation. Representative data from one of two individual experiments are shown (n = 3 per group per experiment). (**B**) IFNγ ELISpot using splenocytes from mice vaccinated with replicons expressing Aatf. Pooled data from two independent experiments (n = 3 or 4 per group per experiment). (**C**) Experimental schematic for prophylactic vaccination in (**C**). Three weeks before tumor-challenge mice are initially vaccinated, replicons are administered intramuscularly (i.m.). Animals are boosted 1 wk before challenge. CBT is administered (i.p.) on days 7, 10, 13, and 16 following subcutaneous (s.c.) implantation of KP-Het[High]. (**D**) KP-Het[High] outgrowth in WT mice. Individual traces for mice dosed with a lipid-only control and treated with CBT or an isotype antibody control, treated only with isotype antibody control and dosed with the replicon or lipid only control, and vaccinated with the replicon and treated with CBT or an isotype control (left to right). Far-right plot is the averaged results. Representative data from three independent experiments (n = 5–10 per group per experiment). *p<0.05, **p<0.01, ***p<0.001; two-way ANOVA (Tukey) in (**A**) and (**D**), Mann–Whitney U in (**B**). Data are shown as mean ± SEM.

The online version of this article includes the following source data for figure 5:

**Source data 1.** Raw data for *Figure 5*.

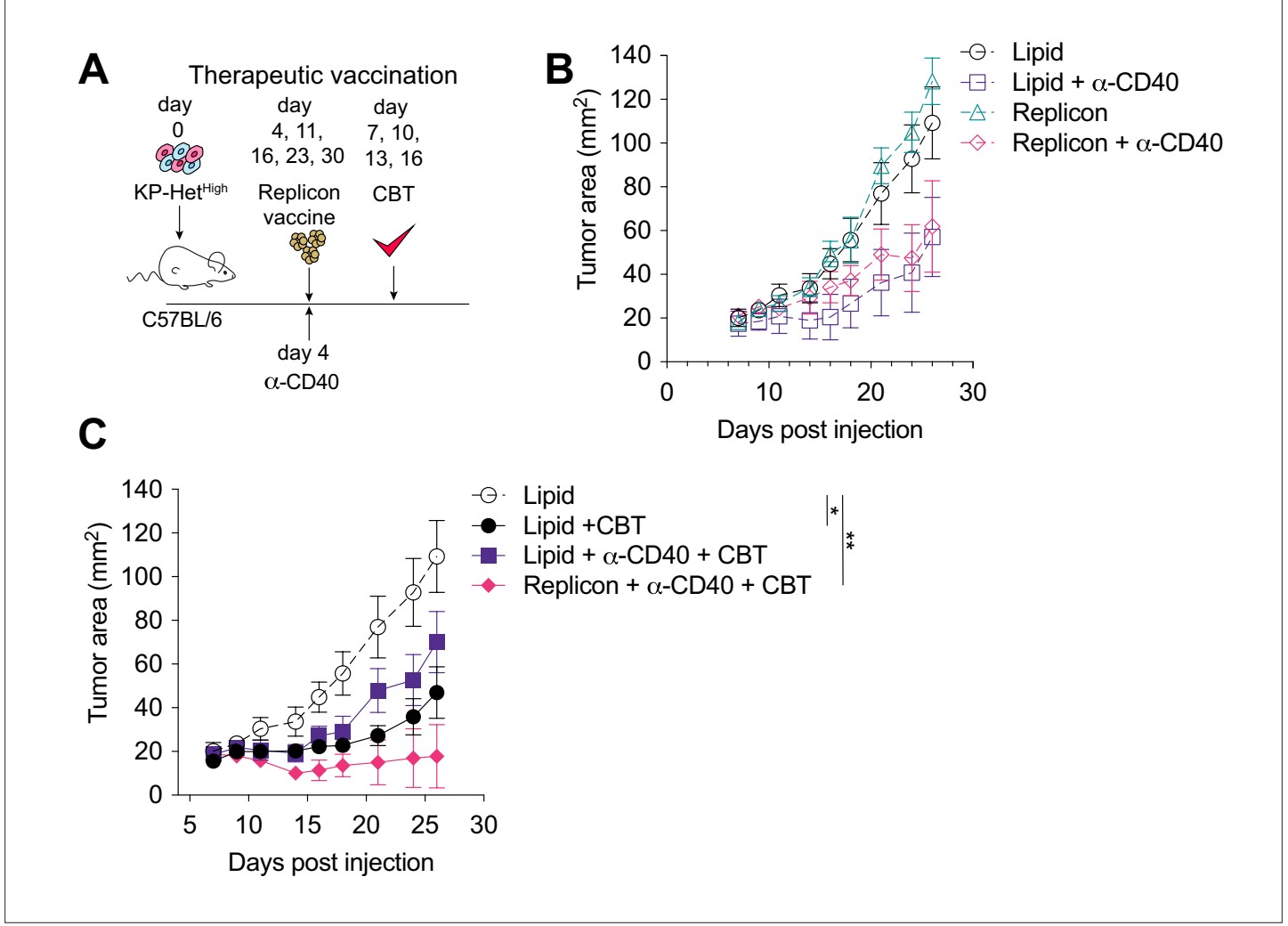

**Figure 6.** Therapeutic mRNA replicon vaccination synergizes with checkpoint blockade immunotherapy (CBT) and CD40 agonism in KP-Het[High] tumors. (**A**) Experimental schematic for therapeutic vaccination in (**B, C**). Animals were implanted with $1 \times 10^6$ KP-Het[High] tumor cells implanted subcutaneously (s.c.). Tumor-bearing mice were vaccinated with replicon vaccine (i.m.) on day 4 post tumor inoculation and continually vaccinated every following week. Anti-CD40 antibody (100 µg) was given with the first vaccination dose only. Dual CBT was administered intraperitoneally (i.p.) on days 7, 10, 13, and 16 post tumor inoculation. (**B**) Tumor growth of KP-Het[High] treated with replicons or lipid only with or without anti-CD40 antibody. (**C**) Tumor growth of KP-Het[High] treated with replicons or lipid only with or without anti-CD40 and CBT. n = 5 per group. The dotted line is the same cohort observed in (**B**) of mice administered with only lipid and isotype control antibodies. *p<0.05, **p<0.01; two-way ANOVA (Tukey) in (**B, C**). Data are shown as mean ± SEM.

The online version of this article includes the following source data for figure 6:

**Source data 1.** Raw data for *Figure 6*.

modest reduction of tumor growth with 1/5 (20%) mice exhibiting complete tumor regression, while CD40 agonism with CBT exhibited mixed response comparable to CBT alone where stable disease was observed in 1/7 (14%) of mice although all tumors eventually progressed (*Figure 6B and C*). These data suggest that for therapeutic vaccination against poorly immunogenic antigens, especially in the heterogeneous setting, optimal priming can be achieved when the vaccine-induced response is augmented with an initial agonistic CD40 treatment and then sustained using CBT treatment.

## Discussion

Clinical and preclinical studies have established that ITH impairs the anti-tumor immune response (*McGranahan et al., 2016*; *Wolf et al., 2019*; *Gejman et al., 2018*), but the mechanism blunting T cell-mediated immunity in tumors with heterogeneous NeoAg expression is still unknown. Understanding

how ITH weakens anti-tumor immunity will enable the development of rational therapies for patients with ITH and increase response rates in patients treated with immunotherapy. We utilized a reductionist approach to study the effect of heterogeneous NeoAg expression on the resulting anti-tumor immune response in a transplantable mouse tumor model. By comparing NeoAgs expressed in subclonal or clonal settings, we discovered that immune responses against poorly immunogenic NeoAg were enhanced when clonally expressed with a strong NeoAg. Mechanistically, we identified that subclonal expression of NeoAgs resulted in presentation of peptides on distinct populations of cross-presenting cDC1, which showed lower expression of costimulatory molecules such as CD40, if presenting only a peptide with a low-binding affinity. Therapeutic vaccination against weak NeoAgs demonstrated synergy with CBT only when the first dose was administered along with an anti-CD40 agonist antibody. In sum, our data suggest that while subclonally expressed NeoAgs elicit weaker anti-tumor T cell responses against less immunostimulatory peptides, these poorly immunogenic peptides may still be viable candidates for therapeutic vaccination in combination with targeted DC maturation.

In tumors with low ITH, we observed migratory cDC1 carrying both antigens to the TdLN. These cDC1 showed higher expression levels of the co-stimulatory ligands CD40 and CD80 compared to cDC1 carrying only the weak NeoAg. It is possible that the difference in tumor debris carried by the cDC1 could result in different interactions with T cells against the stronger NeoAg in the TdLN. Our observation of increased CD40 on cDC1 carrying immunogenic NeoAg is consistent with the concept of DC licensing, where a DC carrying both CD4[+] and CD8[+] epitopes is being 'licensed' by the helper T cell response (**Wu and Murphy, 2022**). Recent work demonstrated that cDC1 can be licensed by CD4[+] T cells, which induce a more mature phenotype via interaction of CD40 with CD40L (**Wu and Murphy, 2022**). By inducing a more mature phenotype, 'licensed' cDC1 were found to have a greater ability to prime the CD8[+] T cell response. Further, it was shown that CD40 expression alone on cDC1 results in a more robust expansion of antigen-specific CD8[+] T cells, further evidence that increased expression of CD40 might directly impact priming of tumor-reactive T cells (**Ferris et al., 2020a**). In our model, cDC1 presenting Adpgk or both antigens expressed more CD40 than cDC1 carrying Aatf debris. This association between CD40 levels and presence of Adpgk on cDC1 suggests that potent CD8[+] T cells could act in a similar manner as CD4[+] helper T cell responses and enhance the stimulatory capacity of DCs. Additional work is needed to delineate transcriptional changes and the timing of these changes on these populations for greater mechanistic insights.

Our data thus far can exclude that the observed effect was primarily driven by epitope spreading, the resulting differences in antigen availability or tumor growth kinetics, as changes in NeoAg-specific T cell responses were preserved following implantation of lethally irradiated tumor cells expressing a single NeoAg or clonal or subclonal NeoAgs in tumors expressing two NeoAgs. While the more immunogenic Adpgk NeoAg response was dependent on the NeoAg load, we observed that the weaker Aatf response was significantly greater in mice implanted with irradiated KP-Het[Low] cells compared to cohorts injected with the same amount of antigen without a strong NeoAg (KP[Aatf]). This result highlights the importance of the context under which a NeoAg is expressed.

For the selection of potent NeoAgs, much emphasis has been placed on the binding affinity of the NeoAg peptide to MHC and the relative binding to its wildtype counterpart (**Schumacher et al., 2019**; **Verdegaal et al., 2016**; **Gubin et al., 2014**; **Duan et al., 2014**). However, these predictions focus on high-affinity NeoAgs in isolation and not on functional immunogenicity. It was recently shown that antigen dominance can dampen anti-tumor immunity in the context of two peptides with similar strong binding affinity (**Burger et al., 2021**). Therapeutic vaccination against the sub-dominant NeoAg enhanced anti-tumor immunity. In our system, however, we observe that the antigen with the higher affinity (Adpgk; 4.3 nM) did provide 'help' rather than competition for binding of H2-Db with the Aatf antigen with a lower affinity of 90 nM. While we did observe that the Adpgk response was dominant, as previously reported (**Kotturi et al., 2008**), we also saw a greater expansion of both NeoAg-specific responses when both antigens were clonally expressed. This observation suggests a therapeutic potential for thus far unused NeoAgs with a lower binding affinity to MHC. It is interesting to note that Aatf was first identified as a potential NeoAg, but was described to be unlikely immunogenic due to its mutant amino acid being located near the carboxy terminus of the peptide, therefore less likely to bind with the TCR (**Yadav et al., 2014**). Our finding underscores the importance of validation of NeoAg immunogenicity in vivo. Furthermore, our study demonstrates the importance of studying NeoAgs with highly variable immunogenicity as this might preclude detrimental effects of

immunodominance between multiple high-affinity NeoAgs. This might be especially important clinically as the NeoAg landscape in patients currently does contain peptides with predicted weaker affinity (*Luksza et al., 2017*).

We developed therapeutic regimens to enhance an Aatf-specific response in KP-Het^High tumors that was comparable to what we observed in KP-Het^Low. This therapy comprised antigen-specific vaccination, anti-CD40 and CBT while neither dual combination was sufficient. This observation suggests that to eradicate a heterogeneous tumor harboring weak and subclonal NeoAgs requires (1) boosting an antigen-specific response, (2) enhancing the productive priming response by providing co-stimulation, and (3) preventing T cell exhaustion by CBT. Our results mirror those from other preclinical models which showed CD40 agonism and CBT synergizing in treating mice with colon tumors (*Westcott et al., 2021*) and significant prolonged survival in a pancreatic cancer model treated with a DC vaccine paired with CD40 agonism (*Lau et al., 2020*). For optimal translation into clinical practice, it will be critical to determine which antigen-specific T cell responses would benefit the most from CD40 agonism and determine how the observed synergy between weak and strong NeoAgs can be exploited best in a vaccine setting. Given the importance of CD4+ T cells in the role to license DC, it will likewise be critical to decipher the differences between CD4+ and CD8+ helper responses, as discussed above. Another consequence of DC licensing is that mature DCs can produce chemokines to facilitate recruitment of naïve CD8+ T cells, thereby increasing the likelihood of interacting with cognate CD8+ T cells to activate (*Castellino et al., 2006*). This notion is highly consistent with our observation that Aatf-reactive T cells are detectable at earlier timepoints when expressed homogeneously compared to the heterogeneous NeoAg expression pattern. Future studies employing tools to detect NeoAg-presentation on cDC1 and identification of NeoAg-reactive T cells in situ will be needed to understand how spatial organization of these cells affects the priming response. While our observations are predominately focused on the initial priming of a NeoAg T cell response, cDC1-mediated re-stimulation of effector T cells is also a critical feature in the TME (*Spranger et al., 2017*; *Gardner et al., 2022*). However, within the tumor the expression patterns of NeoAg will likely be even more critical as spatial analysis of the tumor suggests the formation of areas of clonal growth, resulting in patches of NeoAg expression (*Angelova et al., 2018*; *Milo et al., 2018*). Thus, in tumors with heterogeneous NeoAg expression, cDC1 will consequently only present NeoAg from surrounding tumor cells, thus limiting the stimulatory potential for tumor-infiltrating T cells. It is plausible that this process might accelerate the induction of terminal T cell exhaustion.

In sum, our study provides critical mechanistic insights into how heterogeneous NeoAg expression mediates weaker anti-tumor CD8+ T cell responses. Our work underscores the necessity of improving prediction of functional NeoAg immunogenicity in a patient-specific context, considering ITH, expression level, and binding affinity. Understanding these parameters, their dependencies, and their collective impact on the functional immunogenicity of each NeoAg has the potential to expand the number of actionable NeoAgs for targeted vaccination. The model we developed is a powerful preclinical tool for these future studies with more complex modeling of tumors that better reflect the clinical situation.

## Materials and methods

### Mice

C57BL/6 were purchased from Taconic Biosciences and Jackson Laboratory. *Rag2*^-/- and *Batf3*^-/- mice were purchased from Jackson Laboratory and bred in-house. B6 CD45.1 were purchased from Jackson Laboratory. All mice were housed and bred under specific pathogen-free (SPF) conditions at the Koch Institute for Integrative Cancer Research Building animal facility. For all studies, mice were gender-matched and age-matched to 6–12 weeks old at the start of experiments. All experimental animal procedures were approved by the Committee on Animal Care (CAC/IACUC) at MIT.

### Tumor cell lines and tissue culture

KP1233 and RMA-S were gifts from the Jacks Laboratory at MIT. The KP cell line was validated using SNP analysis, and all cell lines used are routinely tested for mycoplasma. KP6S was subcloned from KP1233. Tumor cell lines were cultured at 37°C and 5% $CO_2$ in culture media (DMEM [Gibco] supplemented with 10% heat-inactivated FBS [Atlanta Biologicals], 1% penicillin/streptomycin [Gibco], and 20 mM HEPES [Gibco]).

## Functional lentiviral titration

KP6S were plated in a 6-well plate to have 20–40% confluency the following day. Lentivirus was serially diluted and cells were transduced as described here. Then, 48 hr later, cells were selected with puromycin or blasticidin (Gibco) and continually selected for 6–8 d until colonies were visible in the wells with the highest dilution. Supernatant was removed and cells were washed once with PBS. 0.5 mL of 10% formalin (Sigma-Aldrich) was added to the wells to fix cells at room temperature (RT) for 5 min. Cells were washed with PBS. 0.5 mL of crystal violet stain (0.05% w/v) was added to stain the cells at RT for 30 min. Cells were washed twice with RODI water. Plates were drained and dried overnight. Blue colonies were counted to determine functional titers.

## Generation of NeoAg-expressing cell lines

KP6S were transduced with the pLV-EF1a-mCherry-mcs lentiviral constructs expressing different NeoAg(s) at an MOI (Multiplicity of Infection) of 0.1. Lentivirus diluted in culture media and supplemented with 4 µg/mL protamine sulfate (Sigma-Aldrich, stock is 1 mg/mL in PBS) was added to cells. The media were changed the following day. Selection began 48 hr after cells were transduced. Flow cytometry was used to confirm and quantify construct expression.

## Predicted binding affinities

Peptide sequences were entered into NetMHC Server 4.0 (https://services.healthtech.dtu.dk/services/NetMHC-4.0/) to determine binding affinities.

## Peptide synthesis

All peptides were synthesized by GenScript at >95% purity with unmodified N- and C-termini.

## RMA-S MHCI stabilization assay

RMA-S cells were cultured in RPMI (Gibco) supplemented with 10% heat-inactivated FBS (Atlanta Biologicals), 1% penicillin/streptomycin (Gibco), and 20 mM HEPES (Gibco) with 55 µM 2-Mercaptoethanol (Gibco). Cells were placed in a tissue culture incubator and incubated at 28°C the day before the experiment. Cells were collected, counted, and resuspended in suspension cell media at $10 \times 10^6$ cells/mL. 100 µL of cells were added to a 96-well plate. A titration of peptide was added to the cells to bring the volume to 200 µL. Cells were incubated for 2 hr at 28°C before antibody staining for flow cytometry analysis.

## Short peptide vaccination

A single dose consisting of 10 µg of peptide (peptide stock is 10 mg/mL resuspended in DMSO) was added to 25 µg cyclic-di-GMP (stock is 1 mg/mL resuspended in PBS) (Invivogen tlrl-nacdg) and PBS added to a final volume of 50 µL. Mice were briefly anesthetized (isoflurane) and the vaccine was administered subcutaneously (s.c.) at the base of the tail. Mice were given a second identical boost 10 d later and spleens were collected 11 d after the boost.

## Tumor outgrowth studies

Tumor cells were collected by trypsinization (Gibco) and washed three times with 1× PBS (Gibco). Cells were resuspended in PBS, and $1 \times 10^6$ tumor cells were injected subcutaneously into the flanks of mice. Subcutaneous tumor area measurements (calculated as length × width) were collected 2–3 times a week using digital calipers until the endpoint of the study.

## DNA extraction and qRT-PCR

DNA was extracted using the Sigma-Aldrich GenElute Mammalian Genomic DNA Miniprep kit following manufacturer's instructions. Extracted DNA was quantified by NanoDrop. DNA was diluted to yield stock concentrations of 50–120 ng/µL. This was further diluted 1:100 or 1:1000 for the reactions. For each plate, a standard was plated using genomic DNA extracted from cell lines along with genomic DNA extracted from tumor tissue. Then, 20 µL reactions were ran [10 µL 2× SYBR Green PCR Master Mix [Applied Biosystems]], 200 µM forward primer, 200 µM reverse primer, 6 µL diluted DNA. Reactions were run on the StepOne Real-Time PCR System (Applied Biosystems), and CT values

were used to determine the amount of DNA contributed by a clone in a single sample. The single-cell suspension implanted into mice on day 0 (Input) was used as a normalization factor:

$$\text{Normalized } KP^{Aatf}/KP^{Adpgk} \text{ mass ratio} = (\text{Sample } KP^{Aatf}/KP^{Adpgk}) / (\text{Input } KP^{Aatf}/KP^{Adpgk})$$

## Tumor dissociation

Tumors were dissected from mice, weighed, and collected in RPMI (Gibco) containing 250 μg/mL Liberase (Sigma-Aldrich) and 1 mg/mL DNase (Sigma-Aldrich). Tumors were minced with dissection scissors or a razor blade and incubated for 45 min at 37°C for enzymatic digestion. Following the digestion, tumor pieces were mashed through a 70 μm filter with a 1 mL syringe plunger to generate a single-cell suspension. The dissociated cells were washed three times with chilled PBS containing 1% heat-inactivated FBS and 2 mM EDTA (Gibco).

## Flow cytometry

Prior to staining, cells were washed with FACS staining buffer (chilled PBS containing 1% FBS and 2 mM EDTA). If cells were used for intracellular staining, Brefeldin A at 1× (BioLegend) was added to all reagents up until the fixation/permeabilization step. Cells were stained for 15 min on ice with eBioscience Fixable Viability Dye eFluor 780 to distinguish live and dead cells and with anti-CD16/CD32 (clone 93, BioLegend) to prevent non-specific antibody binding. Cells were washed once and cell surface proteins were stained for 30 min on ice with fluorophore-conjugated antibodies. Following surface staining, cells were washed twice and analyzed directly or fixed with IC Fixation Buffer (eBioscience) for 20 min at RT for analysis the next day. For intracellular staining, cells were washed twice in wash buffer (eBioscience) and incubated with fluorophore-conjugated antibodies for at least 30 min or overnight at 4°C. Cells were washed twice with FACS staining buffer before running samples. To obtain absolute counts of cells, Precision Count Beads (BioLegend) were added to samples following manufacturer's instructions. All antibodies used are listed in *Supplementary file 1*. Flow cytometry sample acquisition was performed on BD LSRFortessa cytometer and BD Symphony cytometer, and the collected data was analyzed using FlowJo v10.5.3 software (TreeStar).

## Mouse IFNγ-ELISpot

All ELISpot-specific reagents are part of the IFNγ-ELISpot kit from BD Biosciences (Cat# 551083). ELISpot plates were coated overnight at 4°C with anti-IFNγ antibody. Plates were washed and blocked with DMEM supplemented with 10% FBS, 1% penicillin/streptomycin, and 20 mM HEPES for 2 hr at RT. Spleens were harvested from mice and mashed through a 70 μm filter with a 1 mL syringe plunger to generate a single-cell suspension. Red blood cells were lysed with 500 μL of ACK Lysing Buffer (Gibco) on ice for 5 min, and splenocytes were washed three times with chilled PBS. For IFNγ-ELISpot assays using peptide restimulation, $1 \times 10^6$ splenocytes were assayed per well in the presence or absence of 10 μg of peptide. As a positive control, splenocytes were incubated with a mixture of 100 ng/mL PMA (Sigma-Aldrich) and 1 μg/mL ionomycin (Sigma-Aldrich). Following an overnight incubation at 37°C and 5% $CO_2$, plates were developed using the BD mouse IFNγ-ELISpot kit, following manufacturer's protocol.

## Irradiated tumor cell vaccination

Tumor cells were trypsinized, washed once in 1× PBS, passaged through an 18 g needle to generate a single-cell suspension, and further washed twice in 1× PBS. Cells were resuspended at a concentration of $15 \times 10^6$ cells/mL in 1× PBS. Cells were placed in a conical and irradiated with 40 Gy (gray) on ice. Cells were injected into mice immediately following irradiation. Mice were vaccinated as described in short peptide vaccination with both Aatf and Adpgk peptides included in the dose. Spleens were collected for IFNγ-ELISpot on day 21 post implantation of lethally irradiated tumor cells.

## Adoptive T cell transfer

$CD8^+$ T cells were isolated from spleens of naïve or tumor-bearing ($KP^{HetHigh}$ or $KP^{HetLow}$) mice on day 7 after tumor inoculation. $CD8^+$ T cells were enriched using magnetic cell separation ($CD8a^+$ T Cell Isolation Kit, Miltenyi Biotec). Recipient $Rag2^{-/-}$ mice were injected with $5 \times 10^5$ $KP^{Aatf}$ and $KP^{Adpgk}$ tumor

cells on opposite flanks and $5 \times 10^6$ pooled donor CD8$^+$ T cells were transferred retro-orbitally on day 4 after tumor inoculation.

## Immunotherapeutic modulation

For checkpoint blockade therapy, 100 μg each of anti-CTLA-4 (clone UC10-4F10-11, Bio X Cell BP0032) and anti-PD-L1 (clone 10F.9G2, Bio X Cell BP0101) or 100 μg each of isotype controls (clones N/A and LTF2, Bio X Cell BP0091 and BE0090) was administered i.p. diluted in a total volume of 100 μL or 200 μL of PBS on days 7, 10, 13, and 16 following tumor implantation. For CD40 agonism, 100 μg of anti-CD40 (clone FGK4.5, Bio X Cell BP0016-2) was administered i.p. diluted in a total volume of 100 μL of PBS on day 4 following tumor implantation.

## RNA vaccination

Self-replicating RNA based on Venezuelan Equine Encephalitis virus replicons were cloned encoding two copies of the cell-penetrating peptide (CPP) penetratin attached to the previously defined NeoAg Aatf (TCTTTTATGGCTCCAATAGACCATACTACTATGTCAGAT) separated by GGS cleavable sites (CPP-Aatf-CPP-Aatf) under the subgenomic promoter and prepared by in vitro transcription as previously described (*Melo et al., 2019*). Replicons were formulated in lipid nanoparticles by microfluidic nanoprecipitation. The lipids were composed of N1,N3,N5-*tris*(3-(didodecylamino)propyl)benzene-1,3,5-tricarboxamide (TT3) (*Li et al., 2015*), (6Z,9Z,28Z,31Z)-Heptatriaconta-6,9,28,31-tetraen-19-yl 4-(dimethylamino) butanoate (DLin-MC3-DMA; MedChemExpress), 1,2-dioleoyl-sn-glycero-3-phosphoethanolamine (DOPE; Avanti Polar Lipids), Cholesterol (Avanti Polar Lipids), and 1,2-dimyristoyl-*rac*-glycero-3-methoxypolyethylene glycol-2000 (DMG-PEG2k; Avanti Polar Lipids) at a molar ratio of 10:25:20:40:5. RNA (stored in RNAse-free water) was diluted in 10 mM citrate buffer at pH 3.0 (Alfa Aesar). The lipids and RNA were mixed using the NanoAssemblr Ignite instrument (Precision Nanosystems) operated with the following settings: volume ratio 2:1; flow rate 12 mL/min; and waste volume 0 mL. The RNA-loaded LNPs were dialyzed against PBS prior to use. Mice were immunized with 1 μg of replicon RNA in LNPs i.m. in the gastrocnemius muscle.

## Statistical analysis

All statistical analyses were performed using GraphPad Prism (GraphPad).

## Acknowledgements

We thank Melissa Duquette for mouse colony maintenance and laboratory support; and Paul Thompson for administrative support. We thank Leon Yim for technical support. We thank the Koch Institute Swanson Biotechnology Center for technical support. This work was supported by the Melanoma Research Alliance Young Investigator Award, the A Breath of Hope Lung Foundation, the Koch Institute Frontier Research program, and the Ludwig Center at MIT. KBN is supported by the F31 5F31CA261093-02 (NCI) and SS is a Pew-Steward Scholar of the Pew Charitable Trust and holds the Howard S (1953) and Linda B Stern Career Development Professorship.

# Additional information

### Competing interests

Michael E Birnbaum: is an equity holder in 3T Biosciences, and is a co-founder, equity holder, and consultant of Kelonia Therapeutics and Abata Therapeutics. Stefani Spranger: is a SAB member for Related Sciences,Arcus Biosciences, Ankyra Therapeutics and Venn Therapeutics. SS is a co-founder ofDanger Bio. SS is a consultant for TAKEDA, Merck, Tango Therapeutics, Dragonfly andRibon Therapeutics and receives funding for unrelated projects from Leap Therapeutics. The other authors declare that no competing interests exist.

## Funding

| Funder | Grant reference number | Author |
| --- | --- | --- |
| Melanoma Research Alliance | | Stefani Spranger |
| Lung Cancer Research Foundation | | Stefani Spranger |
| National Cancer Institute | F31 5F31CA261093-02 | Kim Bich Nguyen |
| Pew Charitable Trusts | Pew-Steward Scholar | Stefani Spranger |
| Howard S (1953) and Linda B Stern Career Development Professorship | | Stefani Spranger |

The funders had no role in study design, data collection and interpretation, or the decision to submit the work for publication.

## Author contributions

Kim Bich Nguyen, Conceptualization, Data curation, Formal analysis, Investigation, Methodology, Writing – original draft; Malte Roerden, Data curation, Formal analysis, Methodology, Writing – review and editing; Christopher J Copeland, Nory G Klop-Packel, Data curation; Coralie M Backlund, Tanaka Remba, Byungji Kim, Nishant K Singh, Michael E Birnbaum, Darrell J Irvine, Resources; Stefani Spranger, Conceptualization, Formal analysis, Supervision, Funding acquisition, Investigation, Methodology, Writing – original draft, Project administration, Writing – review and editing

## Author ORCIDs

Kim Bich Nguyen http://orcid.org/0000-0002-2269-6809
Christopher J Copeland http://orcid.org/0000-0002-6882-3359
Byungji Kim http://orcid.org/0000-0001-8131-5255
Michael E Birnbaum https://orcid.org/0000-0002-2281-3518
Stefani Spranger https://orcid.org/0000-0003-3257-4546

## Ethics

All experimental animal procedures were approved by the Committee on Animal Care (CAC/IACUC) at MIT.

## Decision letter and Author response

Decision letter https://doi.org/10.7554/eLife.85263.sa1
Author response https://doi.org/10.7554/eLife.85263.sa2

# Additional files

## Supplementary files

- Supplementary file 1. List of antibodies used in the study.
- MDAR checklist

## Data availability

All raw data are uploaded as source data files.

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
