## [Editor Report]

This valuable work explores the influence of intra tumor heterogeneity of neoepitopes within a cancer on the immune response leading to tumor control in vivo using a transplantable murine lung cancer model. It presents convincing evidence that immune responses against weak neoepitopes are enhanced when clonally expressed with strong neoepitopes, due to a more mature DC phenotype and a higher stimulatory capacity of DCs presenting both weak and strong neoepitopes. The work will be of interest to immunologists and cancer immunotherapists.

---

## [Decision Letter]

**Decision letter after peer review:**

Thank you for submitting your article "Decoupled neoantigen cross-presentation in tumors with high intratumor heterogeneity reduces dendritic cell activation to limit anti-tumor immunity" for consideration by *eLife*. Your article has been reviewed by 2 peer reviewers, including Pramod K Srivastava as Reviewing Editor and Reviewer #1, and the evaluation has been overseen by Satyajit Rath as the Senior Editor. The following individual involved in review of your submission has agreed to reveal their identity: Ehsan Ghorani (Reviewer #2).

Essential revisions:

There are some issues that need to be addressed, as outlined below:

1. It is often unclear what the figures show. Figures5 and 6: what are those red and blue circles? Are they HetHigh? Why not say so? What are the dirty yellow cells? These are basic rules of having a legend that tells what is what. A reader can guess, but that should not be the standard of scientific writing.

2. At other times, the rationale of the experiment, the experiment itself, and its interpretation are all unclear. Figure 3E-G are an example. The authors certainly understand their experiments; they just don't communicate it. They summarize these experiments by saying "In sum, we identified that NeoAg expression patterns are critical for priming responses against weak NeoAgs, while the antigen load impacts responses towards strong NeoAgs." Where is the latter part of this summary shown?

3. The manuscript relies of IFNγ production by splenocytes to measure immune responses against defined neoantigens. These measures are used to infer anti-cancer efficacy of reactive clones, but this is potentially problematic. This issue is actually nicely demonstrated by the results presented here: in Figure 1C, KP-Aatf growth is retarded vs KP-control, but in 1D this difference is lost in Rag KO mice……thus suggesting that either an anti-Aatf immune response was partially effective or that KP-Aatf harbours some other targets. By using IFNγ as the major readout and ignoring the fact this may be unreliable to measure anti-cancer efficacy, the reader is left unclear as to whether control of Het-low tumours is related to enhanced activity of anti-Adpgk, anti-Aatf or some other clone. A more convincing/definitive evaluation would be to adoptively transfer splenocytes from tumour bearing mice into animals bearing relevant tumours to show that tumour control can be achieved.

*Reviewer #1 (Recommendations for the authors):*

The manuscript has significant shortcomings.

1. It is often unclear what the figures show. Figures5 and 6: what are those red and blue circles? Are they HetHigh? Why not say so? What are the dirty yellow cells? These are basic rules of having a legend that tells what is what. A reader can guess, but that should not be the standard of scientific writing.

2. At other times, the rationale of the experiment, the experiment itself, and its interpretation are all unclear. Figure 3E-G are an example. The authors must certainly understand their experiments; they just don't communicate it. They summarize these experiments by saying "In sum, we identified that NeoAg expression patterns are critical for priming responses against weak NeoAgs, while the antigen load impacts responses towards strong NeoAgs." Where is the latter part of this summary shown?

3. The Discussion is unhelpfully verbose. Most of it focuses on excessive speculation and hypothesizing and distracts from the important observation of the study and its mechanism. It could be significantly abbreviated and focused. Importantly, far too many citations are careless, in that they overinterpret what has actually been published. These need to be corrected.

4. The title does not convey the message of the paper adequately.

*Reviewer #2 (Recommendations for the authors):*

This is a nice piece of work in many ways, but overall, the manuscript doesn't converge on the question originally posed: why are immune responses less effective against tumours with high ITH? Partly, this arises because the study explores both ITH and immunogenicity. I think the manuscript could be refocussed around the main findings which bear on epitope spreading. Additionally, some of the experimental approaches have important limitations and certain conclusions may be difficult to justify. I hope the following comments are helpful.

1. Cell lines may be different in ways other than neoantigen modification.

It is difficult to be certain that the multiple KP lines developed here are identical other than the modifications described. I agree they appear to have similar growth in Rag2 KO mice (it would be good to show growth of all the cell lines on one plot for comparison), but other differences in immunogenicity/immune evasion may be relevant and under explored. I am reassured that the two versions of Het-low appear to have similar growth characteristics, but note these are not identical! This supports the notion that subclones may harbour uncharacterised differences.

2. Measures of immunogenicity.

The manuscript relies of IFNγ production by splenocytes to measure immune responses against defined neoantigens. These measures are used to infer anti-cancer efficacy of reactive clones, but this is potentially problematic. This issue is actually nicely demonstrated by the results presented here: in Figure 1C, KP-Aatf growth is retarded vs KP-control, but in 1D this difference is lost in Rag KO mice……thus suggesting that either an anti-Aatf immune response was partially effective or that KP-Aatf harbours some other targets.

By using IFNγ as the major readout and ignoring the fact this may be unreliable to measure anti-cancer efficacy, the reader is left unclear as to whether control of Het-low tumours is related to enhanced activity of anti-Adpgk, anti-Aatf or some other clone. A more convincing/definitive evaluation would be to adoptively transfer splenocytes from tumour bearing mice into animals bearing relevant tumours to show that tumour control can be achieved.

3. DC maturation and CD4 responses.

A conclusion presented here is that "CD8^+^ T cell responses against abundant, high affinity antigens can enhance CD8^+^ T cell responses against lower affinity NeoAgs by increasing the stimulatory capacity of cDC1". The suggestion that the CD8 epitope itself is relevant to DC maturation is not safe without considering the alternatives. In particular, the role of CD4 epitopes within the neoantigen sequence needs to be considered/discussed. Several studies have previously shown CD4 responses are frequently obtained from neoantigen vaccines.

4. IFNγ recall, DC maturation and tumour growth dynamics.

Aatf IFNγ recall is enhanced in Het-low vs Het-high bearing mice, but these are shrinking vs growing tumours. Could tumour growth dynamics rather than anti-Aatf responses be relevant here? This goes back to my point above of a more definitive experiments to show Aatf directed anti-cancer effects are enhanced.

A similar point may be relevant to Figure 4D/E: are differences in DC CD40/CD80 expression related to comparison of shrinking vs growing tumours? It isn't fully clear what cell lines were used to generate these data however.

5. Therapy.

The final figures turn to the question of whether immune responses against a subclonal/poorly immunogenic neoantigen can be boosted. But the experiments do not follow clearly on from what the study has established up to this point: that clonal co-expression of Adpgk and Aatf results in stronger immune responses against the latter. Specifically, do the authors suggest that triple therapy (Aatf vaccination/anti-CD40/checkpoint blockade) exploits the antigen spreading effect they have established?

A limitation is that the Het-high model bears no clonal neoantigens; incidentally, this is not representative of human cancers since even the most heterogenous tumours appear to harbour these (for instance, see the 2017 TRACERx NEJM paper). A more realistic model would be to mix Het-low with KP-Adpgk, in order to recreate both clonal (Adpgk) and subclonal (Aatf) neoantigens in a single tumour. The authors could then more cleanly explore their notion that subclonal targeting can be therapeutically beneficial through exploitation of epitope spreading from clonal/strong to subclonal/weakly immunogenic targets.

Finally, it is rather unclear what CD40 agonism is actually doing: does it specifically alter cDC1 biology in this model? It seems necessary to show data on the effects of anti-CD40 on cDC1 biology here. The final figure suggests that vaccine+anti-CD40 is equivalent to anti-CD40 alone: how does this fit with the data presented up to this point? Perhaps testing in a more realistic model of clonal/subclonal mutations will shed light on this.

---

## [Author Response]

Essential revisions:There are some issues that need to be addressed, as outlined below:1. It is often unclear what the figures show. Figures5 and 6: what are those red and blue circles? Are they HetHigh? Why not say so? What are the dirty yellow cells? These are basic rules of having a legend that tells what is what. A reader can guess, but that should not be the standard of scientific writing.

We apologize for the lack of clarity in the figure legends and figures. We have revised the figures to include clear labels on all schematics and have expanded the legends of the figures itself.

To specifically address your questions, the red and blue circles depict KP^Adpgk^ and KP^Aatf^ tumor cells, respectively. In Figure 5 and 6 the mixture of the two indicates that KP-Het^High^ tumors were injected. The yellow circles represent the lipid encapsulated replicon RNA vaccines. Labels have been added directly to the schematics in Figures 5C and 6A along with Figures 1A, 3E and 4A for consistency.

Figure 1:

A) Labeled C57BL/6 mouse and spleen in the schematic.

Figure 3:

E) Labeled tumor cells, C57BL/6 mouse and spleen in the schematic.

Figure 4:

A) Labeled tumor cells, C57BL/6 and lymph node in the schematic.

Figure 5:

C) Labeled vaccine, tumor cells, C57BL/6 and CBT in the schematic.

Figure 6:

A) Labeled vaccine, tumor cells, C57BL/6 and CBT in the schematic.

2. At other times, the rationale of the experiment, the experiment itself, and its interpretation are all unclear. Figure 3E-G are an example. The authors certainly understand their experiments; they just don't communicate it. They summarize these experiments by saying "In sum, we identified that NeoAg expression patterns are critical for priming responses against weak NeoAgs, while the antigen load impacts responses towards strong NeoAgs." Where is the latter part of this summary shown?

We apologize for the ambiguity in the example the reviewer pointed out and for lack of precision. In respect to the example, one conceivable explanation for increased response in KP-Het^Low^ is epitope spreading towards Aatf following an Adpgk response. To ensure that this is not the dominant factor impacting response we lethally irradiated tumor cells either in the KP-Het^High^ or KP-Het^Low^ setting, with irradiated single antigen expressing (KP^Adpgk^ and KP^Aatf^) tumor cells as controls, and assessed immunity. For this experiment we injected the same number of irradiated tumors cells into mice. However, in KP-Het^High^ only 50% of the tumor cells are expressing the Adpgk neoantigen whereas in both KP-Het^Low^ and KP^Adpgk^ 100% of the tumor cells are expressing the Adpgk neoantigen.

What we observed was that debris provided by KP-Het^High^, with only half the tumor cells expressing Adpgk, resulted in an mean of 462 Adpgk-specific T cells/1x10^6^ splenocytes compared to 704.6 and 837.4 Adpgk-specific T cells/1x10^6^ splenocytes when provided with double the number of cells expressing Adpgk with KP-Het^Low^ and KP^Adpgk^, respectively. Thus, we concluded that reduction of neoantigen abundance in KP-Het^High^ tumors resulted in a corresponding reduction in the Adpgk response leading us to conclude that overall antigen load can impact the abundance of the Adpgk-specific T cell responses.

In contrast, this was not observed in the Aatf response. In this case, the weak neoantigen response was dependent on the context of neoantigen expression as irradiated KP-Het^Low^ tumor cells elicited the most robust Aatf response. Given that the synergy between the two T cell responses was preserved in the irradiated KP-Het^Low^ setting we ruled out that tumor cell lysis mediated by Adpgk-specific T cells and thus T-cell dependent epitope spreading as a main driver of increased immunity against KP-Het^Low^ tumors. We have rephrased the text accordingly (lines 454-460) and have carefully revisited other sections for clarity (see yellow highlights in manuscript).

3. The manuscript relies of IFNγ production by splenocytes to measure immune responses against defined neoantigens. These measures are used to infer anti-cancer efficacy of reactive clones, but this is potentially problematic. This issue is actually nicely demonstrated by the results presented here: in Figure 1C, KP-Aatf growth is retarded vs KP-control, but in 1D this difference is lost in Rag KO mice……thus suggesting that either an anti-Aatf immune response was partially effective or that KP-Aatf harbours some other targets. By using IFNγ as the major readout and ignoring the fact this may be unreliable to measure anti-cancer efficacy, the reader is left unclear as to whether control of Het-low tumours is related to enhanced activity of anti-Adpgk, anti-Aatf or some other clone. A more convincing/definitive evaluation would be to adoptively transfer splenocytes from tumour bearing mice into animals bearing relevant tumours to show that tumour control can be achieved.

We thank the reviewers for pointing out this shortcoming of our studies. We agree with the reviewers’ assessment that the addition of Aatf induces some level of immune response and we apologize should we suggest the opposite. Aatf-only cells in fact have a non-zero response as assessed by ELISpot (Figure 2D). However, we observe that this immune response is not sufficient to contribute to a meaningful immune-mediated protection. We have entertained other means of monitoring the effector function of NeoAg responsive T cell responses, such as in vivo killing assays. However, the differences in antigen density due to different binding kinetics make such assays difficult to compare between antigens. Likewise, we had technical difficulties to optimize reliable multimer staining. Hence, we have opted for the ELISpot assay as the most robust assay. We will discuss these points throughout the manuscript to raise awareness.

However, we thank the reviewer for the suggestion to compare immunity by the means of adoptive transfer. We isolated T cell responses from KP-Het^High^ and KP-Het^Low^ tumors on day 7 post tumor implantation and transferred 5x10^6^ T cells into *Rag2^-/-^* mice bearing KP^Aatf^ or KP^Adpgk^ single antigen tumors and assessed tumor growth over time. Indeed, for both responses we observed stronger immune protection by T cells isolated from KP-Het^Low^ tumor bearing mice. These results suggest that the functional quality and abundance of the T cell response is more potent in tumor with homogeneous NeoAg expression compared to heterogenous NeoAg expression.

Reviewer #1 (Recommendations for the authors):The manuscript has significant shortcomings.1. It is often unclear what the figures show. Figures5 and 6: what are those red and blue circles? Are they HetHigh? Why not say so? What are the dirty yellow cells? These are basic rules of having a legend that tells what is what. A reader can guess, but that should not be the standard of scientific writing.

We apologize for the lack of clarity in the figure legends and figures. We have revised the figures to included clear labels on all schematics and have expanded the legends of the figures itself.

To specifically address your questions, the red and blue circles depict KP^Adpgk^ and KP^Aatf^ tumor cells, respectively. In Figure 5 and 6 the mixture of the two indicates that KP-Het^High^ tumors were injected. The yellow circles represent the lipid encapsulated replicon RNA vaccines. Labels have been added directly to the schematics in Figures 5C and 6A along with Figures 1A, 3E and 4A for consistency.

Figure 1:

A) Labeled C57BL/6 mouse and spleen in the schematic.

Figure 3:

E) Labeled tumor cells, C57BL/6 mouse and spleen in the schematic.

Figure 4:

A) Labeled tumor cells, C57BL/6 and lymph node in the schematic.

Figure 5:

C) Labeled vaccine, tumor cells, C57BL/6 and CBT in the schematic.

Figure 6:

A) Labeled vaccine, tumor cells, C57BL/6 and CBT in the schematic.

2. At other times, the rationale of the experiment, the experiment itself, and its interpretation are all unclear. Figure 3E-G are an example. The authors must certainly understand their experiments; they just don't communicate it. They summarize these experiments by saying "In sum, we identified that NeoAg expression patterns are critical for priming responses against weak NeoAgs, while the antigen load impacts responses towards strong NeoAgs." Where is the latter part of this summary shown?

We apologize for the ambiguity in the example the reviewer pointed out and for lack of precision. In respect to the example, one conceivable explanation for increased response in KP-Het^Low^ is epitope spreading towards Aatf following an Adpgk response. To ensure that this is not the dominant factor impacting response we lethally irradiated tumor cells either in the KP-Het^High^ or KP-Het^Low^ setting, with irradiated single antigen expressiong (KP^Adpgk^ and KP^Aatf^) conditions as controls, and assessed immunity. For this experiment we injected the same number of irradiated tumors cells into mice. However, in KP-Het^High^ only 50% of the tumor cells are expressing the Adpgk neoantigen whereas in both KP-Het^Low^ and KP^Adpgk^ 100% of the tumor cells are expressing the Adpgk neoantigen.

What we observed was that debris provided by KP-Het^High^, with only half the tumor cells expressing Adpgk, resulted in an mean of 462 Adpgk-specific T cells/1x10^6^ splenocytes compared to 704.6 and 837.4 Adpgk-specific T cells/1x10^6^ splenocytes when provided with double the number of cells expressing Adpgk with KP-Het^Low^ and KP^Adpgk^, respectively. Thus, we concluded that reduction of neoantigen abundance in KP-Het^High^ tumors resulted in a corresponding reduction in the Adpgk response leading us to conclude that overall antigen load can impact the abundance of the Adpgk-specific T cell responses.

In contrast, this was not observed in the Aatf response. In this case, the weak neoantigen response was depended on the context of neoantigen expression as irradiated KP-Het^Low^ tumor cells elicited the most robust Aatf response. Given that the synergy between the two T cell responses was preserved in the irradiated KP-Het^Low^ setting we ruled out that tumor cell lysis mediated by Adpgk-specific T cells and thus t cell dependent epitope spreading as a main driver of increased immunity against KP-Het^Low^ tumors. We have rephrased the text accordingly (lines 454-460) and have carefully revisited other sections for clarity (see yellow highlights in manuscript).

3. The Discussion is unhelpfully verbose. Most of it focuses on excessive speculation and hypothesizing and distracts from the important observation of the study and its mechanism. It could be significantly abbreviated and focused. Importantly, far too many citations are careless, in that they overinterpret what has actually been published. These need to be corrected.

We thank the reviewer for their critical feedback. We have revised the discussion to increase its focus and reduce overstatements of previously made observations.

4. The title does not convey the message of the paper adequately.

We have changed the title to more accurately describe our findings:

Decoupled neoantigen cross-presentation by dendritic cells limits anti-tumor immunity against tumors with heterogeneous neoantigen expression

Reviewer #2 (Recommendations for the authors):This is a nice piece of work in many ways, but overall, the manuscript doesn't converge on the question originally posed: why are immune responses less effective against tumours with high ITH? Partly, this arises because the study explores both ITH and immunogenicity. I think the manuscript could be refocussed around the main findings which bear on epitope spreading. Additionally, some of the experimental approaches have important limitations and certain conclusions may be difficult to justify. I hope the following comments are helpful.1. Cell lines may be different in ways other than neoantigen modification.It is difficult to be certain that the multiple KP lines developed here are identical other than the modifications described. I agree they appear to have similar growth in Rag2 KO mice (it would be good to show growth of all the cell lines on one plot for comparison), but other differences in immunogenicity/immune evasion may be relevant and under explored. I am reassured that the two versions of Het-low appear to have similar growth characteristics, but note these are not identical! This supports the notion that subclones may harbour uncharacterised differences.

We agree with the reviewer that intraclonal heterogeneity is a strong driving factor impacting anti-tumor immunity. To exclude this feature from our studies we subcloned the KP parental cell line and identified a very stable clone which we called KP6S. We used this subclone to generate all subsequent cell lines used in this study. There may be minor differences in immunogenicity as we cultured the transduced KP6S lines, but we aimed for minimal contribution of tumor cell intrinsic differences affecting the immune response. We apologize if this was not fully clear and we have edited Figure 1B, which includes the schematic of how the lines were generated, the figure legend and the text for clarity (lines 331-333).

2. Measures of immunogenicity.The manuscript relies of IFNγ production by splenocytes to measure immune responses against defined neoantigens. These measures are used to infer anti-cancer efficacy of reactive clones, but this is potentially problematic. This issue is actually nicely demonstrated by the results presented here: in Figure 1C, KP-Aatf growth is retarded vs KP-control, but in 1D this difference is lost in Rag KO mice……thus suggesting that either an anti-Aatf immune response was partially effective or that KP-Aatf harbours some other targets.By using IFNγ as the major readout and ignoring the fact this may be unreliable to measure anti-cancer efficacy, the reader is left unclear as to whether control of Het-low tumours is related to enhanced activity of anti-Adpgk, anti-Aatf or some other clone. A more convincing/definitive evaluation would be to adoptively transfer splenocytes from tumour bearing mice into animals bearing relevant tumours to show that tumour control can be achieved.

We thank the reviewers for pointing out this shortcoming of our studies. We agree with the reviewers’ assessment that the addition of Aatf induces some level of immune response and we apologize should we suggest the opposite. Aatf-only cells in fact have a non-zero response as assessed by ELISpot (Figure 2D). However, we observe that this immune response is not sufficient to contribute to a meaningful immune-mediated protection. We have entertained other means of monitoring the effector function of NeoAg responsive T cell responses, such as in vivo killing assays. However, the differences in antigen density due to different binding kinetics make such assays difficult to compare between antigens. Likewise we had technical difficulties to optimize reliable multimer staining. Hence, we have opted for the ELISpot assay as the most robust assay. We will discuss these points throughout the manuscript to raise awareness.

However, we thank the reviewer for the suggestion to compare immunity by the means of adoptive transfer. We isolated T cell responses from KP-Het^High^ and KP-Het^Low^ tumors on day 7 post tumor implantation and transferred 5x10^6^ T cells into *Rag2^-/-^* mice bearing KP^Aatf^ or KP^Adpgk^ single antigen tumors and assessed tumor growth over time. Indeed for both responses we observed stronger immune protection by T cell isolated from KP-Het^Low^ tumor bearing mice. These results suggest that the functional quality and abundance of the T cell response is more potent in tumors with homogeneous NeoAg expression compared to heterogenous NeoAg expression.

3. DC maturation and CD4 responses.A conclusion presented here is that "CD8^+^ T cell responses against abundant, high affinity antigens can enhance CD8^+^ T cell responses against lower affinity NeoAgs by increasing the stimulatory capacity of cDC1". The suggestion that the CD8 epitope itself is relevant to DC maturation is not safe without considering the alternatives. In particular, the role of CD4 epitopes within the neoantigen sequence needs to be considered/discussed. Several studies have previously shown CD4 responses are frequently obtained from neoantigen vaccines.

We agree with the reviewer that consideration of the CD4 response is critical. We have generated multiple fusion sequences and repeatedly observed the increased immunogenicity of KP-Het^Low^ tumors. The MHC-II epitope prediction analyses varied, identifying potential binders in some constructs and none in others. These are prediction analyses, so we cannot conclude definitively that the constructs themselves do not contain a unique CD4 epitope within the neoantigen sequence. However, we don’t expect these CD4 sequences, if they exist, to be maintained across the various constructs we used in this study, yet we still observe this increased response in the KP-Het^Low^ tumor cell lines.

Furthermore, we regenerated the lines expressing Adpgk and Aatf to express the NeoAgs on separate constructs in order to characterize the dendritic cells draining from the tumor. With these new lines we established tumors that express the same exact sequences whether in KP-Het^High^ or KP-Het^Low^. We expect these two tumors to express the same CD4 epitopes as they have been transduced with identical constructs. The recapitulation of the phenotype that we observed with these new lines (Supplemental Figure 7A-B) suggests that any existing CD4 response, that would be shared by both tumors, is insufficient to contribute to a robust anti-tumor response in KP-Het^High^ tumors.

Thus, for the here shown study we solely focused on MHCI-driven responses. We agree that the established model system would be uniquely positioned to study the contribution of CD4 epitopes to an anti-tumor immune response, yet we believe this is beyond the scope of the work here.

4. IFNγ recall, DC maturation and tumour growth dynamics.Aatf IFNγ recall is enhanced in Het-low vs Het-high bearing mice, but these are shrinking vs growing tumours. Could tumour growth dynamics rather than anti-Aatf responses be relevant here? This goes back to my point above of a more definitive experiments to show Aatf directed anti-cancer effects are enhanced.A similar point may be relevant to Figure 4D/E: are differences in DC CD40/CD80 expression related to comparison of shrinking vs growing tumours? It isn't fully clear what cell lines were used to generate these data however.

We agree with the reviewer that tumor size can contribute to the responses we observe. In order to rule out differences in tumor kinetics contributing to the immune response we injected mice with lethally irradiated KP-Het^High^ and KP-Het^Low^ and assessed immunity. In this case the tumor burden is similar between the two groups before both are completely cleared. Even in this setting where the tumor kinetics are very similar the most robust response is observed in mice injected with irradiated KP-Het^Low^ tumors.

The cell lines used to generate Figure 4D-E used were the new ones referred to in the text in the relevant section, Suppl. Figure 7B. We have modified Figure 4A by labeling the general cell lines used in this study to better match the text. We also edited the figure legend to specify the lines used.

5. Therapy.The final figures turn to the question of whether immune responses against a subclonal/poorly immunogenic neoantigen can be boosted. But the experiments do not follow clearly on from what the study has established up to this point: that clonal co-expression of Adpgk and Aatf results in stronger immune responses against the latter. Specifically, do the authors suggest that triple therapy (Aatf vaccination/anti-CD40/checkpoint blockade) exploits the antigen spreading effect they have established?

As the reviewer points out our data indeed suggest the triple therapy can induce a robust anti-tumor response observed in in mice bearing KP-Het^High^ tumors. Clonal co-expression of both neoantigens does result in a more robust response of both NeoAg-specific T cells and upstream of that, changes to cDC1 maturation. We sought to use these results to develop an effective therapy for KP-Het^High^ tumor-bearing mice. We focused on specifically augmenting the Aatf response by including anti-CD40 with the vaccine. All combined therapies are in clinical trials or approved thus enabling seamless translation of our findings into the clinic.

A limitation is that the Het-high model bears no clonal neoantigens; incidentally, this is not representative of human cancers since even the most heterogenous tumours appear to harbour these (for instance, see the 2017 TRACERx NEJM paper). A more realistic model would be to mix Het-low with KP-Adpgk, in order to recreate both clonal (Adpgk) and subclonal (Aatf) neoantigens in a single tumour. The authors could then more cleanly explore their notion that subclonal targeting can be therapeutically beneficial through exploitation of epitope spreading from clonal/strong to subclonal/weakly immunogenic targets.Finally, it is rather unclear what CD40 agonism is actually doing: does it specifically alter cDC1 biology in this model? It seems necessary to show data on the effects of anti-CD40 on cDC1 biology here. The final figure suggests that vaccine+anti-CD40 is equivalent to anti-CD40 alone: how does this fit with the data presented up to this point? Perhaps testing in a more realistic model of clonal/subclonal mutations will shed light on this.

Thank you for making this point and for the experimental suggestions. Indeed, the clinical cases are likely more complicated than what we have modeled here. The intent of the paper is not to study more subtle changes in ITH as a tumor evolves and the resulting impact on the immune response and vice versa, which is beyond the scope of our study. Rather, we aimed to minimize the impact of other mechanisms of immune evasion in order to study two extremes of ITH as it pertains to NeoAg expression patterns and identify differences in the immune response. The goal of reducing complexity in this model becomes a limitation as the reviewer pointed out.

However, by using this model we were able to identify differences in cDC1 maturation markers between KP-Het^High^ and KP-Het^Low^ tumors. It is still unknown in our model if CD40 engagement is necessary to induce stronger anti-tumor responses and experiments to answer these questions are outside the scope of our work. Instead, we focused on CD40 upregulation in KP-HetLow tumors as an indication of enhanced DC function and a potential dial to manipulate DC function. We targeted CD40 by treating tumor-bearing mice with agonistic anti-CD40 antibodies. We agree that further studies of the downstream impact of CD40 agonism on cDC1 will be critical to better understand if CD40 signaling is necessary or sufficient for optimal anti-tumor responses.

The reviewer also pointed out that vaccine+anti-CD40 is equivalent to anti-CD40 alone. The results are not unexpected. In treatment, when a tumor is already present, raising a tumor-specific response may be insufficient if the tumor reaches a certain size. The additional challenge of T cell exhaustion is especially relevant. The data suggests that broadly targeting the immune response (anti-CD40 alone, anti-CD40+CBT, CBT alone) and specifically directing the immune response toward a certain epitope (replicon alone, anti-CD40+replicon) were both insufficient to see tumor control in KP-Het^High^. Instead, specifically targeting a weakly immunogenic tumor neoantigen and preventing exhaustion of that response was key to tumor control.

We believe our model is a valuable tool that can be modified to study more complex tumors that better match the clinical data. In fact, we have since generated more complex NeoAg setting which match the clinical setting. The obtained immune profiling is highly complex and far beyond the scope of the study shown here, yet the mechanism discovered here of synergy between strong and weak NeoAg responses remains preserved.